# Ubiquitylation-independent activation of Notch signalling by Delta

**Nicole Berndt[1†], Ekaterina Seib[1†], Soya Kim[1,2†], Tobias Troost[1], Marvin Lyga[1], Jessica Langenbach[1], Sebastian Haensch[3], Konstantina Kalodimou[4,5], Christos Delidakis[4,5], Thomas Klein[1]\***

[1]Institute of Genetics, Heinrich-Heine-Universitaet Duesseldorf, Duesseldorf, Germany; [2]Molekulare Zellbiologie, Institut I für Anatomie, Uniklinik Köln, Universität zu Köln, Köln, Germany; [3]Center of Advanced Imaging, Heinrich-Heine-Universitaet Duesseldorf, Duesseldorf, Germany; [4]Institute of Molecular Biology and Biotechnology, Foundation for Research and Technology - Hellas, Heraklion, Greece; [5]Department of Biology, University of Crete, Heraklion, Greece

**Abstract** Ubiquitylation (ubi) by the E3-ligases Mindbomb1 (Mib1) and Neuralized (Neur) is required for activation of the DSL ligands Delta (Dl) and Serrate (Ser) to activate Notch signalling. These ligases transfer ubiquitin to lysines of the ligands' intracellular domains (ICDs), which sends them into an Epsin-dependent endocytic pathway. Here, we have tested the requirement of ubi of Dl for signalling. We found that Dl requires ubi for its full function, but can also signal in two ubi-independent modes, one dependent and one independent of Neur. We identified two neural lateral specification processes where Dl signals in an ubi-independent manner. Neur, which is needed for these processes, was shown to be able to activate Dl in an ubi-independent manner. Our analysis suggests that one important role of DSL protein ubi by Mib1 is their release from cis-inhibitory interactions with Notch, enabling them to trans-activate Notch on adjacent cells.
DOI: https://doi.org/10.7554/eLife.27346.001

**\*For correspondence:**
thomas.klein@hhu.de

[†]These authors contributed equally to this work

## Introduction

Signalling through the Notch signalling pathway is required in many developmental processes of probably all metazoans (*Bray, 2006*; *Fiúza and Arias, 2007*). Notch receptors are hetero-dimeric type 1 trans-membrane proteins, which are activated through ligands of the Delta/Serrate/Lag2 (DSL) protein family. The genome of *Drosophila* contains two DSL ligands, Delta (Dl) and Serrate (Ser), which are structurally related in their extracellular domain (ECD). Their binding elicits two consecutive proteolytic cleavages of Notch performed by Kuzbanian (Kuz)/ADAM10 and the γ-secretase complex, which results in the release of the intracellular domain (NICD) into the cytoplasm. The first, Kuz-mediated cleavage (S2-cleavage) occurs in the extracellular juxta-membrane region and removes the ecto-domain (NECD). The shedding of NECD triggers the second cleavage within the transmembrane domain by the γ-secretase (S3-cleavage), allowing the release of NICD. NICD enters the nucleus and associates with the CSL transcription factor Suppressor of Hairless (Su(H)) and co-factors to activate the transcription of target genes. In the absence of NICD, Su(H) acts as a repressor of transcription in association with Hairless (H) (*Barolo et al., 2002*; *Brou et al., 1994*).

A number of studies have demonstrated the importance of endocytosis for Notch signalling and regulation of its activity (*Le Borgne et al., 2005a*). During signalling, the ligands must be endocytosed in the signal-sending cell to activate Notch in the signal-receiving one. Two membrane-associated E3-ligases, Neuralized (Neur) and Mindbomb1 (Mib1), play a crucial role in the activation of the ligands and their endocytosis (*Le Borgne et al., 2005a*). E3-ligases mediate the ubiquitylation (ubi) of specific substrates at lysines (Ks). This label constitutes a common signal to initiate endocytosis

**eLife digest** Cells use chemical signals to communicate, setting off chains of reactions known as signalling pathways. One key signalling pathway, thought to be required for the development of all animals, is called Notch. In fruit flies, signal proteins known as Delta and Serrate activate the Notch pathway by binding to receptors on the outside of the cell. To do so, the signal proteins first need to be activated themselves.

Two enzymes known as Mindbomb1 and Neuralized activate Delta and Serrate. Both enzymes add a small unit called ubiquitin to specific locations on the signal proteins, but the effect that ubiquitin has on Notch signalling is not yet fully understood.

Berndt, Seib, Kim et al. have now examined fruit flies that had a variety of genetic mutations. These included some flies that could produce mutant versions of the Serrate and Delta proteins that lacked the locations to which ubiquitin normally attaches. The results of the experiments reveal that Delta requires ubiquitin, Mindbomb1 and Neuralized to work at full capacity. However, Delta could still perform some of its roles without ubiquitin. Neuralized and Delta can partner up to send some signals independently of ubiquitin, and Delta can even send some signals on its own. Serrate, on the other hand, does not work at all without ubiquitin.

The results presented by Berndt et al. help us to understand the role that ubiquitin plays in activating Notch signalling. Further work that builds on these findings could help to shed light on how uncontrolled Notch activation can contribute to a variety of diseases, including cancer, cardiovascular diseases and multiple sclerosis.

DOI: https://doi.org/10.7554/eLife.27346.002

and is thought to initiate endocytosis of the DSL ligands. In *Drosophila* the two ligases can functionally substitute for one another in at least some processes such as the wing primordium, suggesting that they perform similar functions, even though they share no obvious sequence similarity (*Weinmaster and Fischer, 2011*). Both ligases contain a Ring Finger domain (RF) that catalyses the ubi.

*neur* is highly expressed in the mesoderm and then all over the neuroectoderm in the embryo and is restricted to neural precursor cells until the pupal stages. Mib1 is ubiquitously expressed, indicating that most DSL signalling is dependent on Mib1. Recent analysis indicates that human Mib1 (MIB1) binds to two peptide motifs in the ICD of the mammalian Ser ortholog Jagged1 (Jag1) ligand termed the N- and C-Box (*McMillan et al., 2015*). Two domains in the Mib1 N-terminus, MZM and REP, are responsible for this binding, with MZM binding the N- and REP the C-box (*McMillan et al., 2015*). Dl has both of these motifs in its ICD; the N-box, also termed ICD2, was shown to be essential for its function (*Daskalaki et al., 2011*). Since MIB1 can substitute for Mib1 in *Drosophila*, it is likely that the interactions between Mib1 and the N-Box are conserved (*McMillan et al., 2015*). Neur binds to a distinct site with the core consensus sequence NXXN, also termed ICD1 in Dl (NEQN in Dl). Such a motif is present in the ICDs of both ligands of *Drosophila* (*Daskalaki et al., 2011*; *Fontana and Posakony, 2009*; *Glittenberg et al., 2006*).

Both ligases are involved in endocytosis of the ligands and their ubi: we recently showed that Neur and Mib1 can ubiquitylate the ICD of Dl in an ICD1 and ICD2 dependent manner, respectively (*Daskalaki et al., 2011*). Additionally, a K at position 742 (K742) of the ICD was identified as the preferred target for Neur in *Drosophila*. Moreover, we found a good correlation between the ability of Dl to undergo ubiquitylation by Neur and Mib1, its internalisation efficiency and its ability to signal. In mammals, a Dll1 variant in which all 17 Ks of its ICD are replaced by arginines (Rs) (Dll1K17) is not ubiquitylated and inactive (*Heuss et al., 2008*). Finally, the *Drosophila* endocytic adapter protein Liquid facets (Lqf), which is the ortholog of Epsin, is absolutely required for the function of DSL ligands (*Overstreet et al., 2004*; *Wang and Struhl, 2004*). The structural analysis of Lqf revealed that its ubiquitin binding motifs (UIMs) are necessary for function (*Overstreet et al., 2004*; *Wang and Struhl, 2004*; *Ho et al., 1989*).

Two, not mutually exclusive, models have been suggested to explain why the activity of the ligands requires endocytosis. In the first model (pulling force model) the Epsin-mediated entry of ubiquitylated ligands in Clathrin-coated pits creates a pulling force that is essential for shedding of

the NECD. In the second model (recycling model), ligand signalling activity is thought to depend on events taking place after internalisation of the ligands into endosomes, such as enzymatic processing of the ligand into the active form or packaging into exosomes: accordingly, signalling activity would require recycling of the ligands to the cell surface to be able to activate Notch. Previous work has provided evidence for both models (reviewed in [*Weinmaster and Fischer, 2011*]).

Although Mib1 and Neur are clearly required for DSL signalling, their role is not entirely understood: (1) While endocytosis of Ser is virtually abolished in *mib1* mutant cells, that of Dl is not obviously affected, although both Ser and Dl function is impaired (*Wang and Struhl, 2004*). (2) Recent work suggests that Neur might have a function that is separable from its ligase function and required for endocytosis (*Skwarek et al., 2007*). (3) The phenotype of *mib1* mutants is milder than that of mutants of other genes involved in Notch signalling. These results raise the possibility that one or both ligands might possess an activity that is independent of ubi.

In addition to activating Notch in adjacent cells (trans-activation), the ligands engage in inhibitory interaction with Notch molecules present in the same cell (*Klein et al., 1997*; *Micchelli et al., 1997*). This cis-inhibition occurs if DSL ligands are expressed above a threshold level. The underlying mechanism is not well understood, but cis-inhibition can be suppressed by co-expression of Notch, indicating that the ratio between Notch and ligand concentration in a cell is an important parameter (*Klein et al., 1997*). Moreover, it has been shown that the ECD of the ligands is involved (*Glittenberg et al., 2006*). Whether the ICD is involved, or cis-inhibition is influenced by Mib1 and Neur is not established, although there is evidence that enhanced/diminished ligand endocytosis relieves/aggravates (respectively) cis-inhibition of Ser and Dl (*Glittenberg et al., 2006*)

Here, we tested the activity of a Dl variant (termed DlK2R) in which all 12 Ks of its ICD were replaced by structurally related arginines (Rs). We show that DlK2R possesses signalling activity. However, its activity is reduced compared to Dl, indicating that ubi is required for the full function of Dl. The reduction in activity is in part caused by increased cis-inhibitory activity of DlK2R indicating that the ICD and its ubi contribute to the degree/strength of cis-inhibition of a ligand. We found that an important function of Mib1 during normal signalling is the release of the ligands from cis-inhibitory interaction with Notch. Our results revealed that Dl can signal in two ubi-independent modes, one dependent and one independent of Neur. The results also indicate that Neur and Mib1 have different mechanisms to activate Dl. In contrast to Mib1, Neur can promote Dl signalling in an ubi-independent manner. We identified neuron sibling specification during larval neurogenesis and also specification of the sensory organ precursor of the bristle sensillum as naturally occurring instances of Dl/Notch signalling, where Dl requires Neur, but ubi seems to be dispensable. Finally, we found an instance of Dl induced Notch signalling that is independent of both Neur and ubi.

## Results

### *mib1* mutants exhibit residual Notch pathway activity, which depends on Dl

During wing development of *Drosophila* interactions between dorsal and ventral cells mediated by the Notch pathway at the dorso-ventral (D/V) compartment boundary result in the establishment and maintenance of a stripe of expression of Wingless (Wg) that straddles the D/V boundary of the wing disc at the end of the third larval instar stage (*Figure 1A*, arrow) (*Klein, 2001*). During the initial phase, dorsal boundary cells signal to ventral boundary cells to activate Wg expression and to increase expression of Dl (*Micchelli et al., 1997*; *de Celis and Bray, 1997*; *de Celis et al., 1996*; *Troost and Klein, 2012*). The ventral boundary cells signal back through Dl to activate expression of Wg in dorsal cells (*Figure 1B*, step 1). After establishment, expression of Wg in boundary cells is maintained through a feedback loop (DS-loop, *Figure 1B* steps 2 and 3): it involves the induction of expression of Dl and Ser in cells adjacent to the boundary cells by Wg signalling (*Figure 1B* step 2). Wg expression at the D/V boundary is then maintained by back signalling from these cells to boundary cells through the Notch ligands (*Figure 1B* step 3). Loss of the activity of Notch results in the loss of expression of Wg along the D/V boundary and a reduction of the diameter of its ring-like domains of expression in the proximal wing. The reduction depends on the degree of loss of Notch pathway activity. Strong loss of activity, as observed in mutants of the γ-secretase component *Psn*, reduces the inner ring-like domain to a spot or causes its complete absence (*Figure 1C*, arrowhead)

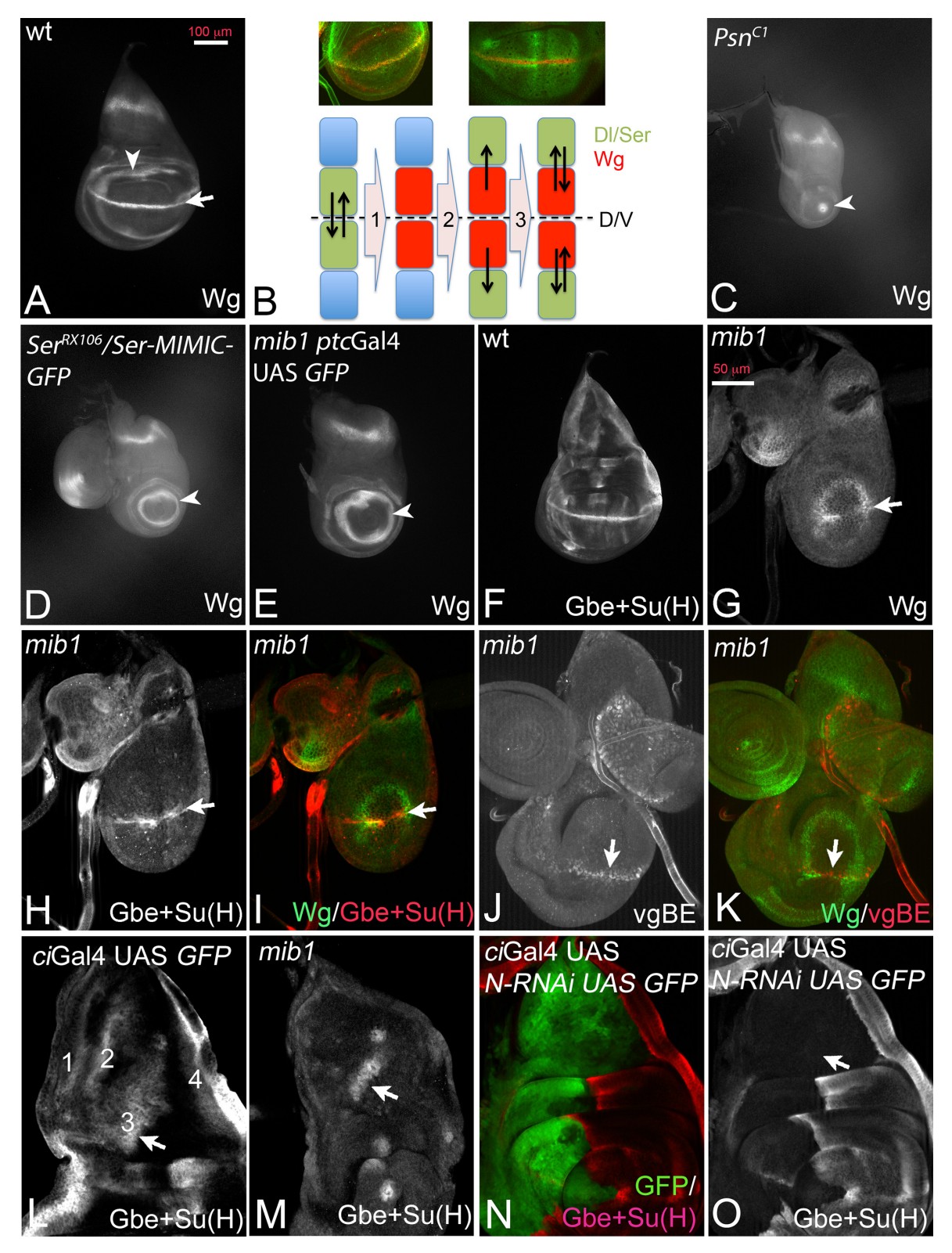

**Figure 1.** Dl-induced activity of Notch in *mib1* mutant wing imaginal discs. (**A**) Expression of Wg in a normal wing disc. (**B**) Establishment of the Wg expression domain along the DV boundary by Notch signalling. For further explanation, see text. (**C–E**) Expression of Wg in mutants of genes involved in Notch signalling. (**C**) *Psn-*, (**D**) *Ser-* and (**E**) *mib1*-mutant wing imaginal disc. The arrow in (**A**) points to the expression along the D/V boundary that is dependent on Notch signalling. It is lost in the mutants. The arrowhead in (**A, C–E**) points to the inner ring-like expression domain of Wg. Its diameter

*Figure 1 continued on next page*

*Figure 1 continued*

is a measure of the loss of Notch pathway activity. (**F**) Expression of Gbe + Su(H) in a wildtype disc. (**G–I**) Expression of Wg (**G, I**), Gbe + Su(H) (**H, I**) in a *mib1* mutant disc of the early third instar stage. The arrow highlights the expression along the D/V boundary. (**J, K**) Expression of *vg*BE (**J, K**) and Wg (**K**) in a *mib1* mutant disc of the early third instar stage. The arrow points to the expression of *vg*BE and Wg along the D/V boundary. (**L–O**) Expression of Gbe + Su(H) in the notum of a wildtype (**L**), *mib1* (**M**), and Notch depleted disc (**N, O**). The arrow highlights the S3 expression domain of Gbe + Su (H). S3 is present in *mib1* mutants (**M**, arrow), but absent in *Notch* depleted nota (**O**, the arrow points to the expected location of S3).

DOI: https://doi.org/10.7554/eLife.27346.003

The following figure supplement is available for figure 1:

**Figure supplement 1.** Residual Dl induced Notch activity in *mib1* mutant wing imaginal discs.

DOI: https://doi.org/10.7554/eLife.27346.004

(*Klein and Martinez-Arias, 1998*; *Koelzer and Klein, 2006*; *Struhl and Greenwald, 1999a*; *Ye et al., 1999*). Weaker loss of Notch activity, as observed in *Ser* mutants (*Klein and Martinez-Arias, 1998*), causes a reduction in diameter of the domains (*Figure 1D*, arrowhead).

The phenotype of *mib1* null mutant wing discs resembles that of *Ser* mutants (*Figure 1E*, arrowhead; compare with D). We previously showed that residual activity of Dl is responsible for the weaker phenotype of *Ser* mutants (*Klein and Martinez-Arias, 1999*). The resemblance prompted us to search for residual Notch pathway activity in *mib1* null mutants. We found that the Notch pathway is indeed weakly active. The Notch activity reporter Gbe + Su(H) was initially expressed along the D/V boundary in early third larval instar stages of *mib1* mutant wing discs and lost only later in development (*Figure 1F–I*, arrow). Also the *vestigial* boundary enhancer (*vg*BE) (*Williams et al., 1994*), a direct endogenous target of the Notch pathway, was expressed in early wing discs (*Figure 1J,K*, arrow).

In the notum Gbe + Su(H) is expressed in four stripes (*Figure 1L*). We observed residual expression of stripe 3 in *mib1* mutants (S3; *Figure 1L,M*, arrow). S3 expression is dependent on Notch signalling, since depletion of the activity of Notch by expression of *N*-RNAi in the notum with *ci*Gal4 resulted in the complete abolishment of its expression (*Figure 1N,O*, arrow in O). Together, these results confirm that residual ligand-dependent Notch activity is present in *mib1* mutants.

*Hairless* (*H*) encodes a member of the transcriptional repressor complex assembled around Su(H) in the absence of Notch activity to repress expression of the target genes of the pathway (*Barolo et al., 2002*). Loss of its function results in de-repression of a subset of target genes and enhancement of residual Notch signalling in *Ser* mutants, which is sufficient to re-establish the expression of Wg and Gbe + Su(H) along the D/V boundary of the wing disc (*Troost and Klein, 2012*; *Klein et al., 2000*). We observed that, as in the case of *Ser* mutants, concomitant loss of *H* function partly restores the expression of Wg and maintains that of Gbe + Su(H) along the D/V boundary in *mib1* mutants (*Figure 1—figure supplement 1A–C*, arrow). The comparison with the expression of the dorsal selector Apterous (Ap) indicated that the expression of Gbe + Su(H) was restricted to dorsal boundary cells (*Figure 1—figure supplement 1D–F*). Since Ser is unable to activate the Notch pathway in dorsal cells because of the activity of Fng (*Panin et al., 1997*), this observation strongly suggests that Dl signalling is responsible for the re-appearance of the expression of the Notch target genes in *mib1 H* double mutants.

The results indicate that a residual activity of the Notch pathway, probably induced by Dl, is present in the absence of *mib1* function. Since Mib1 is the only known E3-ligase for Notch ligands known to be involved during wing development, the results raise the possibility that Dl has an activity that is independent of ubi. Although it has been shown that Mib1 is absent in the discs mutant for the *mib1* null alleles used here (*Le Borgne et al., 2005*), it might be possible that a long-lasting maternal component of *mib1* exists that provides cells of the imaginal disc with residual Mib1 activity. It is also possible that another unidentified E3-ligase contributes to Dl signalling. These issues are addressed in the following.

## Activation of the Notch pathway by Delta in the absence of *mib1*

In the experiments described in the following, we expressed a set of UAS *Dl* constructs with *ptc*Gal4 to test their activity in the wing imaginal disc. If not stated otherwise, the constructs are inserted in the attP landing site 51C (Flybase: M{3xP3-RFP.attP'}ZH-51C) to achieve similar expression levels and allow direct comparison (*Bischof et al., 2007*). All constructs are HA-tagged at their C-terminus.

*ptc*Gal4 is expressed on the anterior side of the anterior-posterior boundary (A/P boundary) in a gradient that increases towards the posterior. Its posterior expression boundary coincides with the A/P boundary (*Figure 2A–C*, arrow in B). Continuous expression of Dl-HA with *ptc*Gal4 at 25°C induces ectopic expression of the Notch target gene Wg and the more sensitive Notch activity reporter Gbe + Su(H) in two stripes perpendicular to their normal expression along the D/Vboundary, as has been reported for untagged versions (*Doherty et al., 1996*) (*Figure 2D–F*, white arrows in D, E). The anterior stripe (a in *Figure 2E*) is located in the anterior region of low expression, while the posterior stripe (p in *Figure 2E*) is adjacent to the *ptc* domain in posterior non-expressing boundary cells (*Figure 2F*). Wg is not induced in regions of high Dl expression close to the A/P boundary (*Figure 2D,E*, yellow arrow, F), because of cis-inhibition of Notch by high Dl concentrations (*Doherty et al., 1996*). The cis-inhibition also interrupts expression of endogenous Wg along the D/V boundary (*Figure 2D,E*, white and yellow arrows, F).

If Dl is active in *mib1* mutants, ectopic expression of Dl-HA should induce ectopic activation of Notch in this mutant (*Figure 2G–J*). Indeed, expression of Dl-HA in *mib1* mutants caused weak and diffuse ectopic expression of Wg in the *mib1* mutant discs (100% penetrance, n = 7; *Figure 2I*, arrow). In addition, the diameter of the ring-like domains of Wg is increased (*Figure 2I*, arrowhead, compare with H, arrowhead). Expression of the more sensitive Gbe + Su(H) was induced throughout the *ptc*Gal4 expression domain, even in the notum where, in contrast to the wing, Notch activity does not induce expression of the ligands (*Figure 2J*, arrow and arrowhead). Dl-HA was not able to induce ectopic expression of the activity reporters in *Psn* mutants, in which the pathway is interrupted on the signal-receiving side (*Figure 2—figure supplement 1A–C*). To rule out that residual activity of Mib1 supports Dl signalling in the mutants, we generated a Dl variant where the core of the Mib1 binding box in its ICD (ICD2, see *Figure 3A*, [*Daskalaki et al., 2011*]) is mutated to alanine (Dl-HA$^{i2\ ala}$). This variant induced activity of the Notch pathway in the *mib1* mutant background in a manner comparable to Dl-HA throughout the *ptc*Gal4 domain (*Figure 2K,L*, arrow and arrowhead, compare with I, J). This finding indicates that Dl can activate the Notch pathway in a *mib1* independent manner and strongly argues against a residual Mib1 activity in the mutants.

The expression of a Ser-HA variant, also inserted into the 51C landing site, failed to induce ectopic expression of Wg and Gbe + Su(H) in *mib1* mutants, indicating that it is completely dependent on Mib1 (*Figure 2M,N*).

Altogether, the results confirm on the one hand a strong dependency of Dl on the function of *mib1*. On the other hand, they indicate that Dl, in contrast to Ser, can signal in the absence of the function of *mib1* and raise the possibility that part of the activity of Dl is independent of ubi. Alternatively, it is possible that the residual activity of Dl is induced by the activity of an unidentified Dl specific E3-ligase.

## Dl with a K-free ICD can signal in the absence of the function of *mib1*

To discriminate between the possibilities, we generated a variant of Dl where all 12 Ks of its ICD are replaced by the structurally similar arginine (R) (DlK2R-HA, *Figure 3A*) and inserted it into the genomic 51C landing site. It has been shown that a corresponding variant of Dll1 is not ubiquitylated (*Heuss et al., 2008*). In agreement, we here found that, in contrast to Dl-HA, DlK2R-HA is not ubiquitylated by Mib1 or Neur in S2 cells (*Figure 3—figure supplement 1*).

Expression of DlK2R-HA with *ptc*Gal4 resulted in the interruption of Wg expression along the D/V boundary suggesting a strong negative effect on Notch signalling at the D/V boundary (*Figure 3B*, yellow arrow). The width of Wg interruption was much larger than that caused by Dl-HA suggesting that DlK2R-HA has a significantly stronger cis-inhibitory activity compared to Dl-HA (*Figure 3B* compare with *Figure 2E,* yellow arrow). In an attempt to quantify the increase in cis-inhibition, we measured the width of the gap induced in the expression domain of Wg by expression of Dl-HA and DlK2R-HA (*Figure 3—figure supplement 2A–C*). We found that the gap induced by Dl-HA was on average 3,3 (n = 10) nuclei and that induced by DlK2R-HA 16,7 nuclei (n = 9). This dramatic increase in the gap size of Wg expression confirms the strong increase in the cis-inhibitory abilities of DlK2R-HA.

However, DlK2R-HA was not completely inactive. It induced ectopic expression of Wg in posterior boundary cells in 44% of discs (n = 9; *Figure 3E*, arrows), suggesting that it still possesses signalling activity. Indeed, we found that DlK2R-HA consistently induced ectopic expression of the more sensitive Gbe + Su(H) reporter in two stripes throughout the *ptc*Gal4 domain (*Figure 3C, D*, arrowhead).

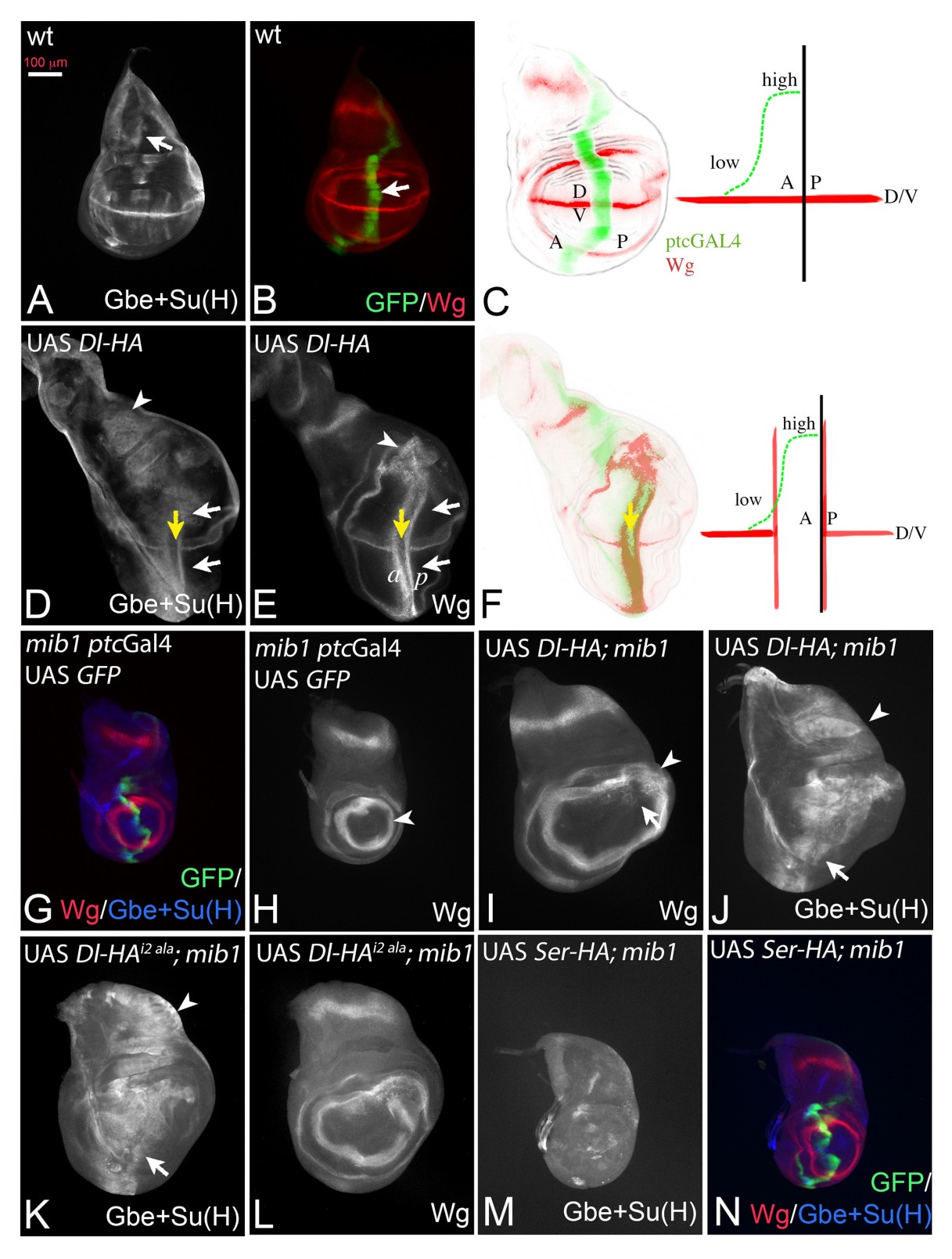

**Figure 2.** Dl induced activity in *mib1* mutant wing imaginal discs. (**A, B**) Normal expression patterns of Gbe + Su(H) (**A**), Wg and *ptc* Gal4 (**B**). The arrow in (**A**) points to the S3 expression of Gbe + Su(H) and in (**B**) to the expression domain of *ptc*Gal4. (**C**) The expression of *ptc*Gal4 occurs in a gradient that increases from anterior until the A/P compartment boundary. It is not expressed in posterior cells and occurs perpendicular to the expression of Wg along the D/V boundary. A: anterior; P: posterior; D: dorsal; V ventral. (**D, E**) Expression of Dl-HA in a wildtype disc results in ectopic activation of

*Figure 2 continued on next page*

*Figure 2 continued*

Wg (**E**, white arrows) and Gbe + Su(H)-lacZ (**D**, white arrows). The yellow arrow in (**D, E**) points to the intersection of *ptc*Gal4 with the D/V boundary where the expression of Wg and Gbe + Su(H) is interrupted due to the cis-inhibitory effect of high Dl expression. Note that Gbe + Su(H) is ectopically induced throughout the whole *ptc*Gal4 domain, which also runs through the notal region (arrowhead in **D**). In contrast the ectopic expression domain of Wg is restricted to the wing area and stops at the inner ring of Wg expression (arrowhead in **E**). (**F**) One stripe of the ectopic expression domains of Wg is located in regions of low expression of the anterior compartment. The second stripe is induced in posterior boundary cells that do not express Dl-HA by trans-signalling from anterior boundary cells. (**G, H**) Expression of *ptc*Gal4, Wg and Gbe + Su(H) in *mib1* mutant wing imaginal discs. The arrowhead in (**H**) points to the inner ring of Wg expression. (**I, J**) Expression of Dl-HA in a *mib1* mutant disc. Weak ectopic expression of Wg (**I**, arrow) is observed . The ring-like domains of Wg expression are expanded in comparison to *mib1* mutants (arrowhead, compare with H). (**J**) Ectopic expression of Gbe + Su (H) is induced throughout the *ptc*Gal4 domain (arrow and arrowhead). (**K–L**) Expression of UAS *Dl*$^{i2\ ala}$-HA in *mib1* mutant discs results in a similar phenotype as expression of Dl-HA (compare with I, J). (**M, N**) Expression of Ser-HA is not able to induce expression of Gbe + Su(H) (**M**) or Wg (**N**) in *mib1* mutant wing discs.

DOI: https://doi.org/10.7554/eLife.27346.005

The following figure supplement is available for figure 2:

**Figure supplement 1.** Dl-HA cannot induce activation of the Notch pathway in *Psn* mutant wing imaginal dioscs.

DOI: https://doi.org/10.7554/eLife.27346.006

DlK2R-HA was not able to induce expression of Gbe + Su(H) in *Psn* mutant discs, indicating that its expression depends on the activity of the Notch pathway (***Figure 3F***). Importantly, DlK2R-HA was able to induce strong expression of Gbe + Su(H) in *mib1* null mutants, indicating that it can activate the Notch pathway in a *mib1* independent manner (***Figure 3G–J***). Also weak ectopic expression of Wg was induced in a fraction of *mib1* discs (18%, n = 17; ***Figure 3I***, arrow). The activity of DlK2R-HA was independent of the presence of the Neur binding site in its ICD, as a variant where the Neur binding site (NEQNAV, see (***Fontana and Posakony, 2009***); termed i1 in ***Figure 3A***) is mutated to alanines (DlK2R$^{i1\ ala}$-HA) induces the expression of Gbe + Su(H) in *mib1* mutants in a manner comparable to DlK2R-HA (***Figure 3K,L***, arrowheads, compare with G). This finding excludes the possibility that so far undetected weak ubiquitous expression of Neur activates DlK2R-HA in *mib1* mutants.

In an earlier round of constructs UAS *DlK2R-HA* had been randomly inserted into the genome and a second- and third-chromosomal insertion had been selected for analysis. These constructs behaved in the same manner as the 51C insertion, but the phenotypes were more severe (***Figure 3—figure supplement 3***). Expression of each insertion by *ptc*Gal4 resulted in a splitting of the wing primordium into two small halves (***Figure 3—figure supplement 3A–D***, yellow arrow in A, B, D). A similar split was observed if a dominant-negative variant of Dl, Dl$^{stu}$, or a Notch-RNAi construct is expressed (***Figure 3—figure supplement 3E–H***, yellow arrow in E, G). This similarity further confirms that the loss of the Ks strongly increases the cis-inhibitory abilities of Dl. In contrast to Notch-RNAi and Dl$^{stu}$, DlK2R-HA induced an ectopic stripe of Wg expression in the adjacent posterior boundary cells, indicating that it can activate Notch in trans in non-expressing neighbours (***Figure 3—figure supplement 3A–C***, arrowheads in A, B, arrow in C). The phenotype of DlK2R-HA expression could be mimicked by co-expression of N-RNAi and Dl in wing imaginal discs (***Figure 3—figure supplement 3I,J***, compare with A, B) confirming the increase of cis-inhibitiory abilities without a loss of trans-signalling for the K free DlK2R-HA.

We sought to suppress the strong cis-inhibitory effect of DlK2R-HA. In the case of Dl-HA, this had been achieved through co-expression with Notch (***Klein et al., 1997***; ***Doherty et al., 1996***) (***Figure 3M***, yellow arrow, compare with ***Figure 2E***). When co-expressed with Notch, DlK2R-HA (in 51C) produced the same phenotype. It induced a band of strong ectopic expression of Wg and Gbe + Su(H) and the endogenous expression of Wg along the D/V boundary was not interrupted (***Figure 3N–P***, yellow arrow in N). This finding shows that the ability of DlK2R-HA to activate the Notch pathway is partly obscured by its increased cis-inhibitory abilities. The co-expression of Dl-HA and DlK2R-HA with Notch in *mib1* mutants resulted in the induction of a broad stripe of expression of Gbe + Su(H) (***Figure 3Q–T***). However, induction of Wg expression was severely reduced or absent. Although we did not examine the mechanistic basis of this defect in depth, a possible explanation for the reduction of activity of the Dl variants in *mib1* mutants is the loss of the wing specific DS-loop. In line with this notion, we found that the ability of Dl-HA and DlK2R-HA to induce expression of endogenous Dl in *mib1* mutant discs was severely reduced (***Figure 3—figure supplement***

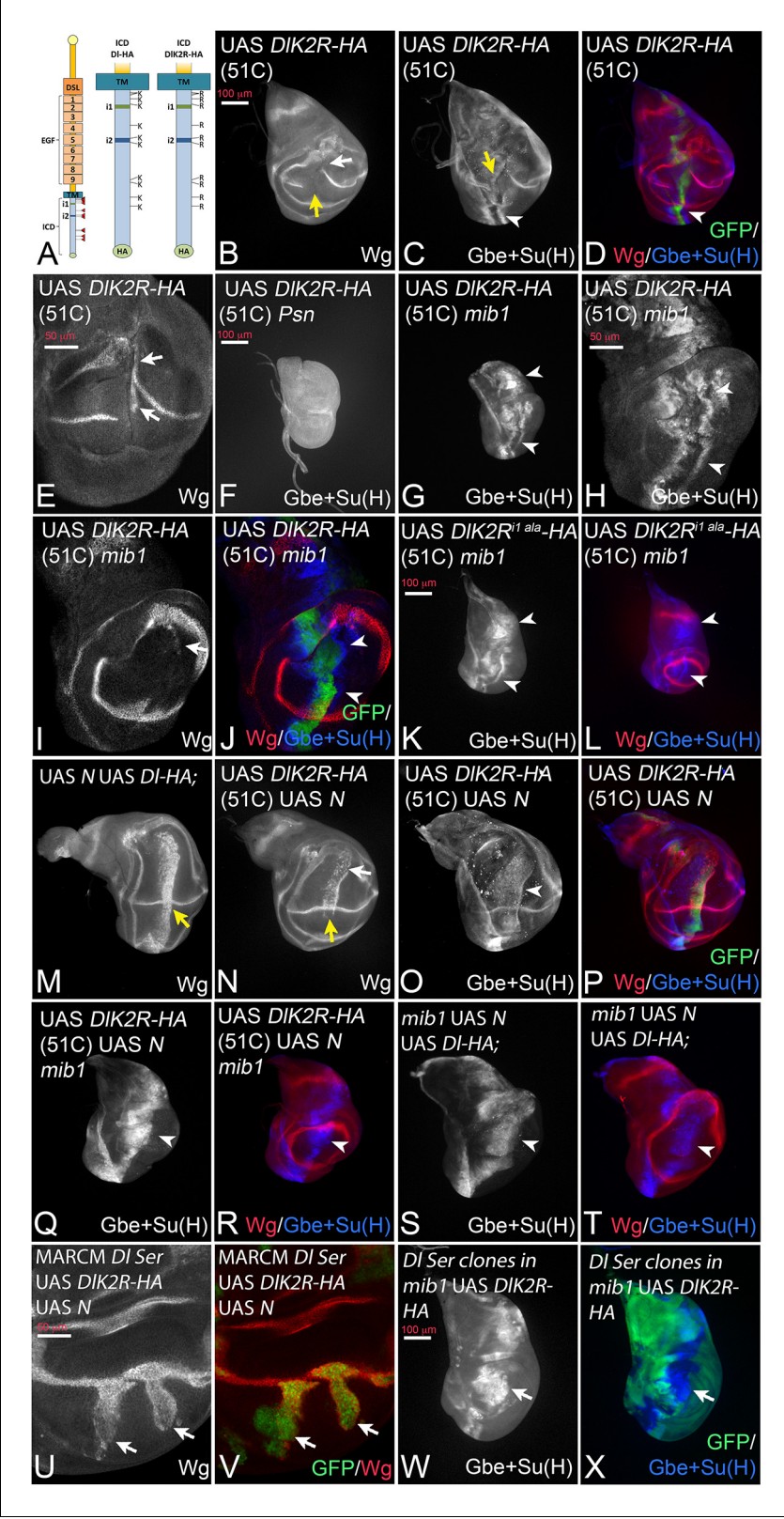

**Figure 3.** Expression of DlK2R-HA (inserted into the landing site 51C) with *ptc*Gal4. (**A**) Schematic representation of Dl-HA and DlK2R-HA and the localisation of the i1 and i2 binding boxes for Neur and Mib1 respectively in the ICD of Dl. (**B–D**). Expression in wildtype discs interrupts the expression of Wg and of Gbe + Su(H) at the D/V boundary (**B**, **C**, yellow arrow). Expression Gbe + Su(H) is ectopically induced (**C**, **D**, arrowhead). (**E**) Ectopic

*Figure 3 continued on next page*

*Figure 3 continued*

expression of Wg (arrows) is observed in a fraction of the discs. (F) Expression of DlK2R-HA in *Psn* mutant discs does not induce Gbe + Su(H). (G–J) Expression of DlK2R-HA in *mib1* mutant discs. (H–J) Magnification of the wing area of the disc shown in (G). The expression of DlK2R-HA causes the induction of two parallel stripes of Gbe + Su (H) (arrowheads in H, J) along the *ptc* domain. Wg expression is only weakly induced (I, arrow). (K, L) Expression of DlK2R$^{i1\ ala}$-HA results in a similar induction of Gbe + Su(H) expression as DlK2R-HA (arrowheads). (M) Co-expression of Dl-HA with Notch results in the induction of a broad stripe of expression of Wg in the wing area. Moreover, no gap in the expression domain of Wg along the D/V boundary is observed (yellow arrow, compare with *Figure 2E*). (N–P) A similar suppression of the increased cis-inhibitory effect of DlK2R-HA was observed upon co-expression with Notch (yellow arrow in N). (Q, R) Co-expression of DlK2R-HA with Notch in a *mib1* mutant disc, reliably induces a broad stripe of Gbe + Su(H) expression (arrowhead). (S, T) Co-expression of Notch and Dl-HA in *mib1* mutants. A broad band of expression of Gbe + Su(H) is induced throughout the *ptc*Gal4 domain (arrowhead). (U, V) *Dl Ser* double mutant MARCM clones co-expressing DlK2R-HA and Notch. Wg is induced throughout the clone in the absence of endogenous ligands (arrows). (W, X) A *mib1* mutant disc that bears *Dl Ser* double mutant clones (loss of GFP) and expresses DlK2R-HA with *ptc*Gal4. DlK2R-HA is able to induce expression of Gbe + Su(H) in the *Ser Dl* double mutant cells (arrow).

DOI: https://doi.org/10.7554/eLife.27346.007

The following source data and figure supplements are available for figure 3:

**Figure supplement 1.** Ubiquitylation of Dl-HA or DlK2R-HA in S2 cells by Neur and Mib1.

DOI: https://doi.org/10.7554/eLife.27346.008

**Figure supplement 2.** Quantification of the gap in the endogenous expression of Wg along the D/V boundary induced by cis-inhibition of the Dl constructs.

DOI: https://doi.org/10.7554/eLife.27346.009

**Figure supplement 2—source data 1.** Quantification of cis-inhibition of Dl variants.

DOI: https://doi.org/10.7554/eLife.27346.012

**Figure supplement 3.** The consequences of expression of a randomly inserted DlK2R-HA construct with *ptc*Gal4.

DOI: https://doi.org/10.7554/eLife.27346.010

**Figure supplement 4.** Induction of Dl expression by the Dl variants.

DOI: https://doi.org/10.7554/eLife.27346.011

*4A–G*). Therefore, most of the Notch activity in *mib1* mutants is generated by expression of the exogenous Dl constructs. It is probably too weak to consistently activate the expression of Wg.

Using the MARCM technique, we co-expressed DlK2R-HA with Notch in *Dl Ser* double mutant cells and found that it induced expression of Wg in the mutant cells (*Figure 3U,V*; arrows). Moreover, we expressed DlK2R-HA in *mib1* mutant wing discs and found that the activity marker Gbe + Su(H) was induced also in cells of *Dl Ser* double mutant clones (*Figure 3W,X*, arrow). These results indicate that the activation of the Notch pathway by DlK2R-HA is independent of the endogenous ligands.

In summary, the results confirm that DlK2R-HA has a residual activity despite the loss of all Ks in its ICD. At least part of this activity is independent of *mib1*, and probably ubi-independent.

## DlK2R-HA displays enhanced stability

In order to test whether DlK2R-HA is endocytosed, we monitored its sub-cellular distribution in comparison to the endosomal marker Rab7-YFP, which marks maturing endosomes (MEs) (*Figure 4A–F*; see also M and M). Dl-HA was found at the apical membrane (*Figure 4A*, arrowhead) and in Rab7-positive MEs (*Figure 4A–C*, arrow), as previously described. We found that DlK2R-HA was distributed similarly to Dl-HA at the apical membrane (*Figure 4D*, arrowhead) and in Rab7 positive MEs (*Figure 4D–F*, arrow) and localised with similar frequency at Rab7 positive MEs as Dl-HA (*Figure 4G*). Thus, DlK2R-HA is endocytosed into approximately the same number of endosomes. This is in agreement with previous findings, which show that endocytosis of Dl is not affected in a *mib1* mutant background (*Le Borgne et al., 2005*).

Although the overall distribution in endosomes may not be affected by *mib1* activity, we had earlier detected significant changes in endocytic rate of Dl upon compromising the intracellular motifs mediating Mib1 interaction (*Daskalaki et al., 2011*). We therefore decided to assay possible stability differences between Dl-HA and DlK2R-HA. For this purpose, we ubiquitously expressed Dl-HA and DlK2R-HA for a limited time with *tub*Gal4 *tub*Gal80$^{ts}$ and evaluated their relative concentrations

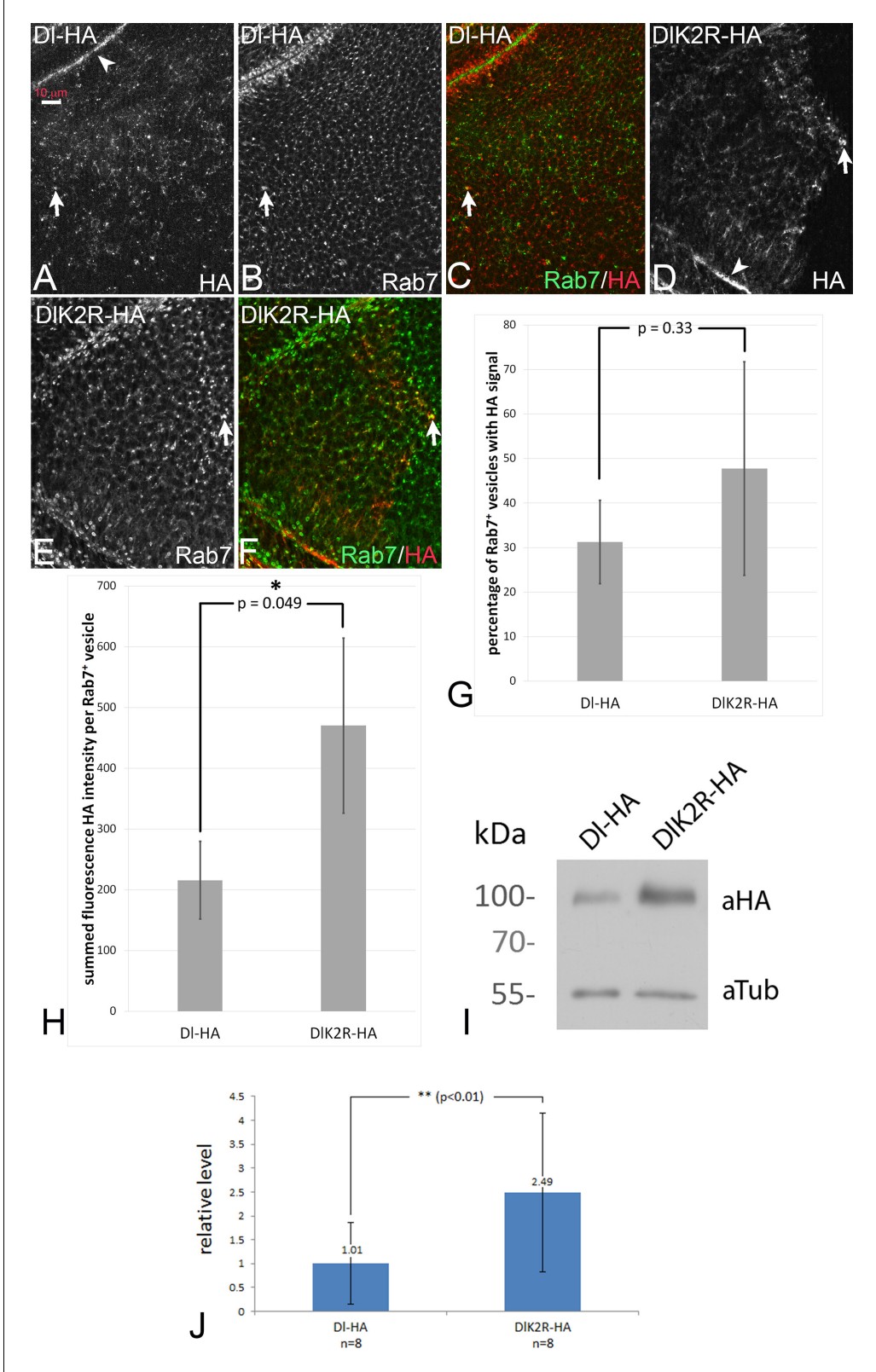

**Figure 4.** Endocytosis of Dl-HA and DlK2R-HA. (**A–C**) Subcellular localisation of Dl-HA and DlK2R-HA (**D–F**) in wing imaginal disc cells. The variants are detected at the apical plasma membrane (arrowhead in A, (**D**) and in Rab7 positive MEs (arrows). (**G**) Quantification of the percentage of Rab7 positive endosomes that are also positive for the Dl variants. (**H**) Summed fluorescence HA intensity of Rab7 positive vesicles in discs that either express UAS Dl-HA or DlK2R-HA expressed with *ci*Gal4. For further information of the method, see M and M. The intensity is significantly higher for DlK2R-HA
*Figure 4 continued on next page*

*Figure 4 continued*

expressing discs. (I) Representative Western blot of the expression of Dl-HA and DlK2R-HA with *tub*Gal4 Gal80[ts] for 48 hr, revealing stronger expression of DlK2R-HA. (J) Quantification of the expression of the Dl variants. The average value of ratio between the band of the Dl variant and the Tubulin loading control from eight independent Western blots was determined. DlK2R-HA is expressed approximately 2.5x more than Dl-HA. The P-value was calculated with a Student's t-test.

DOI: https://doi.org/10.7554/eLife.27346.013

The following source data and figure supplement are available for figure 4:

**Source data 1.** Raw median data of cargo intensity measurements and raw percentage data of colocalization analysis.

DOI: https://doi.org/10.7554/eLife.27346.015

**Source data 2.** Quantifications of Western blots.

DOI: https://doi.org/10.7554/eLife.27346.016

**Figure supplement 1.** Degradation of Dl-HA and DlK2R-HA in the wing imaginal disc.

DOI: https://doi.org/10.7554/eLife.27346.014

with Western blot analysis (eight independent blots). This analysis revealed that DlK2R-HA is expressed at an approximately 2.5 x higher level than Dl-HA (*Figure 4I,J*). Thus, the Ks in the ICD have a detectable negative effect on the stability of Dl (*Daskalaki et al., 2011*).

To pinpoint the phase in the endosomal pathway where the delay in degradation of DlK2R-HA is caused, we measured the fluorescence intensity of the HA staining of the Rab7 positive MEs shown in *Figure 4A–F* (see M and M for details). We found that the summed fluorescence intensity in the MEs was significantly higher in discs that expressed DlK2R-HA compared to discs that expressed Dl-HA (*Figure 4H*). This suggests that DlK2R-HA accumulates in MEs and is less efficiently transported to the lysosome than Dl-HA.

Next, we expressed Dl-HA and DlK2R-HA in a pulse-chase experiment using a combination of *ci*Gal4 and *tub*Gal80[ts], and measured the time required for their degradation. After 16 hr of expression all cells of the anterior compartment reliably expressed either Dl-HA or DlK2R-HA (*Figure 4—figure supplement 1A–E*). After a chase of 24 hr, Dl-HA was completely degraded (*Figure 4—figure supplement 1C,D*). At this time point, we still observed high DlK2R-HA levels, also at the plasma membrane (*Figure 4—figure supplement 1G,H*, arrows in H). Low levels of DlK2R-HA were observed even after 36 hr of chase (*Figure 4—figure supplement 1I,J*). These results further confirm that the degradation of DlK2R-HA is delayed and indicate that at least two effects cause this delay: (1) Inefficient endocytosis and (2) Less efficient transport to the lysosome after endocytosis. Altogether, the results are consistent with the documented requirement of ubi at several steps in the endosomal pathway.

## A Dl variant that cannot bind to Mib1 and Neur possesses residual activity and displays increased cis-inhibition

Dli1/2 is a randomly inserted variant that lacks the binding sites for Neur (ICD1) and Mib1 (ICD2) (*Figure 3A*). As a consequence, it is not ubiquitylated by these E3-ligases (*Daskalaki et al., 2011*). Based on the expression of Wg, the expression of Dli1/2 with *ptc*Gal4 caused a phenotype that resembled that of DlK2R-HA, including the increased cis-inhibition judged from the large gap inflicted on the endogenous Wg stripe (*Figure 5A*, compare with *Figure 3B*). The quantification of the gap in Wg expression confirmed the increase in cis-inhibition (*Figure 3—figure supplement 2*). This resemblance combined with our results raised the possibility that also Dli1/2 has residual activity, although a direct comparison of the two variants is not possible due to the unknown genomic location of Dli1/2. We monitored the induction of the more sensitive Gbe + Su(H) upon expression of Dli1/2. Indeed, we observed induction of ectopic expression of Gbe + Su(H) in wildtype, as well as in *mib1* mutant wing discs (*Figure 5B–F*, arrowhead in B, arrow and arrowhead in E). Moreover, its co-expression with Notch induced a broad stripe of ectopic expression of Wg and Gbe + Su(H) (*Figure 5G–I*, arrow and arrowhead in G, H). Likewise, co-expression of Dli1/2 with Notch in *mib1* mutant discs strongly induces expression of Gbe + Su(H) (*Figure 5K,L*), but not of Wg (*Figure 5J,L*). The activation of Wg expression achieved through co-expression of Notch and Dli1/2 in the wildtype is independent of the endogenous ligands, since it is also observed in *Dl Ser* double mutant MARCM clones (*Figure 5M,N*, arrow). The observed phenotypes are similar to those of expression of DlK2R-

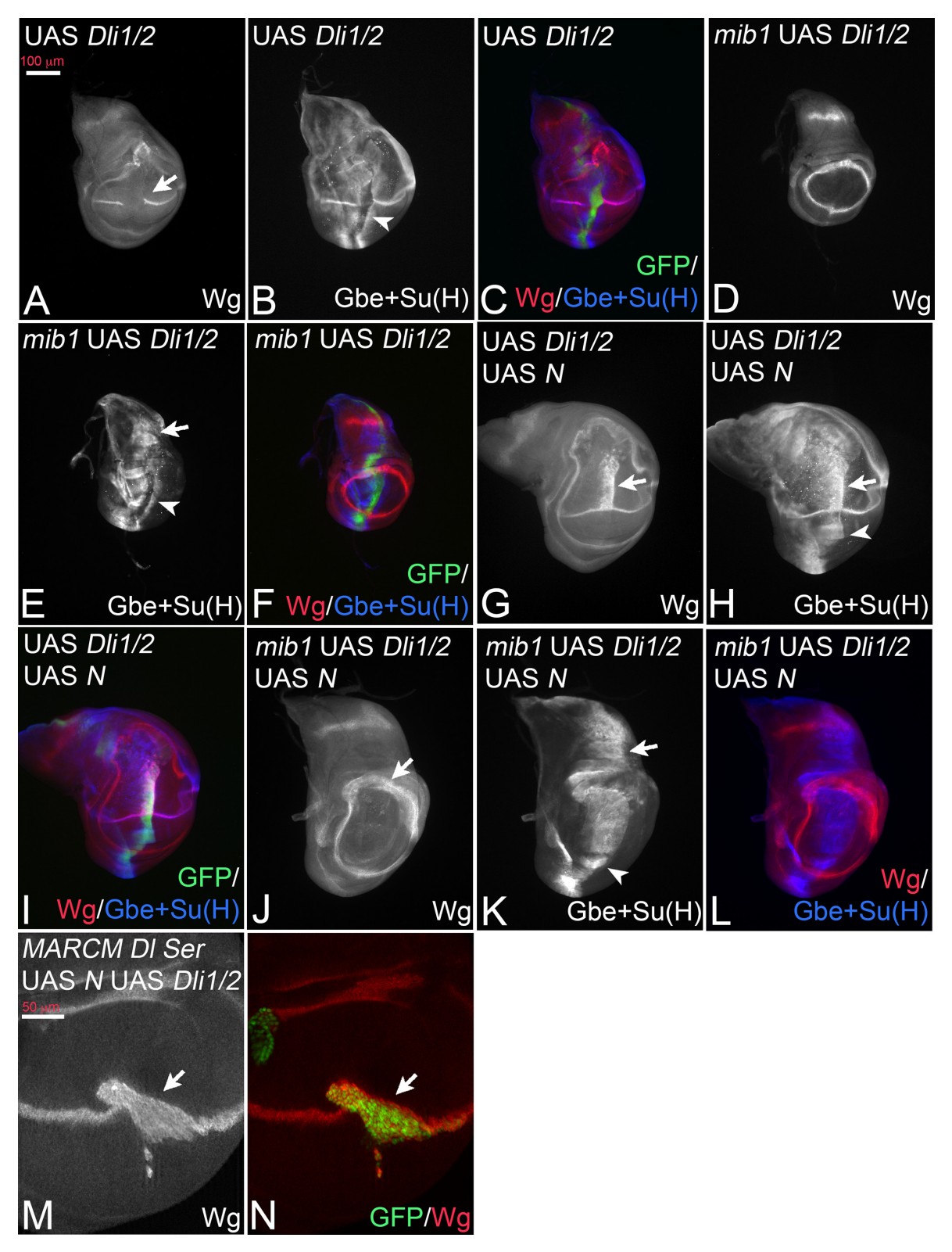

**Figure 5.** Expression of Dli1/2 with *ptc*Gal4. (**A–C**) Expression in wildtype discs causes an interruption of the expression of Wg along the D/V boundary at the point of intersection with the *ptc* domain (A, arrow) and induces ectopic expression of Gbe + Su(H) (B, arrowhead). (**D–F**) Expression of Dli1/2 in *mib1* mutant discs fails to induce Wg expression (**D**), but induces strong expression of Gbe + Su(H) throughout the *ptc*Gal4 domain (E, arrow and arrowhead). (**G–I**) Co-expression of Dli1/2 with Notch in the wildtype results in the induction of a broad stripe of ectopic expression of Wg (G, arrow)
*Figure 5 continued on next page*

*Figure 5 continued*

and of Gbe + Su(H) (**H**, arrow and arrowhead) in a similar manner as co-expression of Notch with DlK2R-HA (Compare with *Figure 3N-P*, *Figure 2J–L*, ). (**J–L**) Co-expression of Dli1/2 with Notch in *mib1* mutant results in the expansion of the ring-like expression domains of Wg (**J**, arrow) and ectopic expression of Gbe +Su(H) throughout the *ptc*Gal4 domain (**K**, arrow and arrowhead). (**M, N**) *Dl Ser* double mutant MARCM clones (arrow, green in **N**) that co-express Dli1/2 with Notch. The mutant cells express Wg.

DOI: https://doi.org/10.7554/eLife.27346.017

HA (compare with *Figure 3*). If another E3-ligase could ubiquitylate Dli1/2, we would expect Dli1/2 to signal more strongly than DlK2R-HA, since we know that ubi enhances the activity of Dl.

The similarity of the phenotypes of Dli1/2 and DlK2R-HA suggests that no other E3-ligase is involved in Dl/Notch signalling in the wing disc. The results further support the notion that Dl has a residual activity that is independent of Mib1 and Dl-ICD ubi. This ubi-independent activity is likely to cause the weaker than expected *mib1* mutant phenotype. The increase in cis-inhibition observed for DlK2R-HA and Dli1/2 also uncovers a so far little appreciated involvement of ubi of the ICD of Dl in this process. Thus, Mib1 might be required to overcome cis-inhibition through ubi of the Dl-ICD.

## Relief of cis-inhibition initiates ligand-dependent signalling in *mib1* mutant wing imaginal discs

The results so far suggest that the loss of Notch activity in *mib1* mutants may result from excessive cis-inhibition caused by endogenous levels of the DSL ligands. Cis-inhibition is determined by the ratio between the concentrations of DSL ligands versus Notch (*Klein et al., 1997*), implying a mechanism whereby DSL ligands and Notch expressed in the same cell engage in a non-productive stoichiometric complex (see *Figure 6A*). To determine the extent to which Notch signalling could be augmented in *mib1* discs by relieving cis-inhibition, we made several manipulations that imbalance the stoichiometry between Notch and its ligands.

We induced *Dl Ser* double mutant clones in wildtype and *mib1* mutant wing discs. It has been shown that the loss of function of *Dl* and *Ser* in cells of the wing pouch and the eye results in the activation of the Notch pathway by relieving cis-inhibition, when the mutant cells are located adjacent to ligand expressing wildtype cells (*Micchelli and Blair, 1999*; *Miller et al., 2009*) (*Figure 6A*). Dl and Ser are widely expressed in the notum and in combination their patterns cover most of the notal area (*Troost et al., 2015*) (*Figure 6—figure supplement 1A–D*). Analysis of *Dl Ser* mutant clones revealed that cis-inhibition occurs also throughout a large part of the notum of normal wing discs: we observed activation of the Notch pathway in mutant boundary cells, indicated by the ectopic activation of Gbe + Su(H) (*Figure 6B–D*, arrowhead in C, D). Thus, cis-inhibition is not restricted to wing and eye development, but appears to be a frequently used mechanism to regulate Notch activity.

We next tested whether relief of cis-inhibition can also be observed in *mib1* mutant discs. Indeed, strong activation of Gbe + Su(H) in mutant cells at the boundary of *Dl Ser* clones was observed in the notum, indicating that the loss of the ligands can result in strong activation of the Notch pathway also in the absence of *mib1* function (*Figure 6E–J*, arrows). As expected, activation occurred only in areas where the ligands were expressed (*Figure 6G–I*). Note, that this Notch activity in *Dl Ser* mutant cells is induced by the activity of ligands in adjacent ligand-expressing cells, although these are devoid of the function of *mib1* function. Since Ser is not able to activate the Notch pathway in the absence of *mib1* function, even when over-expressed (see *Figure 2M,N*), the signalling must be mediated by Dl. Thus, Dl can signal in the absence of *mib1* function without being over-expressed and ubiquitylated. The results also show that at least part of the loss of Notch activity in *mib1* mutants is caused by cis-inhibition through the globally expressed ligands and supports the conclusion that Mib1 is required to overcome cis-inhibition during normal signalling.

The experiments so far were performed with the artificial reporter construct Gbe + Su(H). To confirm the results also with the promoter of an endogenous target gene, we monitored the expression of E(spl)mß-lacZ, which is expressed in a similar global manner as Gbe + Su(H) (*Cooper et al., 2000*) (*Figure 6—figure supplement 1E–G*). E(spl)mß behaved like Gbe + Su(H), as its expression was initiated in mutant boundary cells of *Dl Ser* double mutant clones in *mib1* mutant discs (*Figure 6—figure supplement 1I–L*, arrowhead).

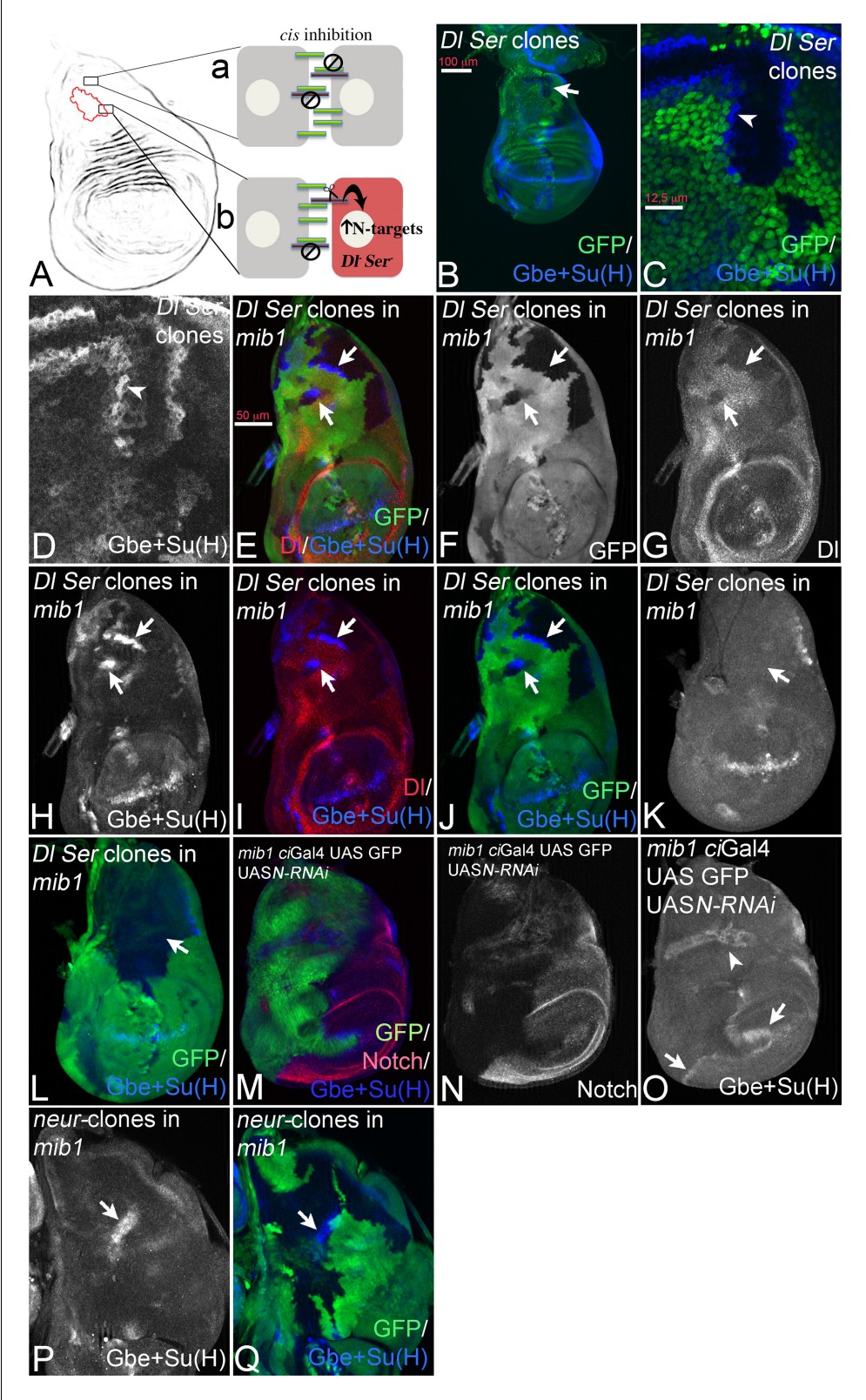

**Figure 6.** Analysis of cis-inhibition in *mib1* null mutant wing discs in the early third instar stage. (**A**) Relief of cis-inhibition. In several areas of the notum of the wing disc cis-inhibition prevents significant Notch signalling (**a**). (**b**) Cells that are located at both sides of the boundary of a clone double mutant for *Ser* and *Dl*. Mutant cells have lost their ligands and therefore the Notch receptor is no longer engaged in cis-inhibitory interactions and can be activated by the ligands of the adjacent Dl and Ser positive cells on the other side of the boundary. Consequently the Notch pathway is activated in the

*Figure 6 continued on next page*

*Figure 6 continued*
mutant cells, which consequently activate the expression of the Notch target genes. (B–D) Cis-inhibition occurs in the notum. *Dl Ser* double mutant clones in the notum of a wildtype wing imaginal disc. (C, D) Magnification of the area with a clone highlighted by the arrow in (B). The clone is labelled by the absence of GFP. Expression of Gbe + Su(H) is induced in the mutant cells at the clone boundary because of relief of cis-inhibition ( arrowhead). (E–J) Expression of Gbe + Su(H) is also induced in *Dl Ser* double mutant cells at the boundary of *Dl Ser* double mutant clones in *mib1* mutant cells in the notal area (arrows). Thus, ligand dependent Notch activation caused by relief of cis-inhibition also occurs in the absence of *mib1* function. (K, L) A large *Dl Ser* double mutant clone that covers most of the notal area of a *mib1* mutant disc. The S3 expression of Gbe + Su(H) normally present in *mib1* mutant discs is lost. The arrows point to the expected region of S3 expression (compare with *Figure 1M*, arrow). (M–O) Depletion of *Notch* function in the anterior compartment of a wing disc by expression of *Notch*-RNAi with *ci*Gal4. It causes the induction of Gbe + Su(H) expression in the posterior cells adjacent to the *ci*Gal4 expression domain (O, arrows). The arrowhead in (O) points to the unrelated expression of Gbe + Su(H) in the trachea, which is attached to the imaginal disc, but is no part of it. (P, Q) Loss of Neur function does not affect the S3 expression of Gbe + Su(H) in *mib1* mutant discs. *neur* mutant clones are labelled by the absence of GFP. The arrow points to S3 expression in a clone.

DOI: https://doi.org/10.7554/eLife.27346.018

The following figure supplement is available for figure 6:

**Figure supplement 1.** Activation of E(spl)mß expression in *mib1* discs through relief of cis-inhibition.
DOI: https://doi.org/10.7554/eLife.27346.019

Cis-inhibition is mutual: the non-productive cis DSL-Notch complex inhibits the activity of both receptor and ligand (*Becam et al., 2010*). We therefore next tested whether removing Notch expression in a group of cells would release the ligands from cis-inhibition and would result in trans-activation of the Notch pathway in adjacent cells. We created this situation by expression of a *Notch*-RNAi construct that efficiently depletes Notch expression if expressed with *ci*Gal4 in the anterior compartment (*Figure 6M–O*). In this situation, anterior cells that lack Notch were exposed to posterior Notch expressing cells at the A/P boundary. Consequently, the endogenous ligands of the anterior cells released from cis-inhibition should signal to posterior boundary cells. Indeed, this was observed in *mib1* mutant wing imaginal discs: The expression of Gbe +Su(H) was induced in adjacent posterior boundary cells (*Figure 6O*, arrows). This result confirms that endogenous Dl can signal in the absence of *mib1* function. It also indicates that Dl, not engaged in cis-interaction, can to some extent out-compete cis-interacting ligands in adjacent cells for interaction with Notch.

## An instance of Dl/Notch signalling that is independent of Mib1 and Neur

In *mib1* mutant wing discs residual Gbe + Su(H) expression (S3) is observed (*Figure 1L*, arrow). S3 expression is lost in *mib1* discs if the notum lacks expression of the ligands in this area, indicating that S3 is induced by Dl-dependent Notch signalling (*Figure 6K,L*, arrow). However, expression of S3 in *mib1* mutant discs is independent of Neur, as it is still detected in *neur* mutant clones (*Figure 6P,Q*). We observed a S3-like residual expression domain of E(spl)mß-lacZ in *mib1* mutants (*Figure 6—figure supplement 1H, J*, arrow). The depletion of Notch with *Notch*-RNAi also abolished the expression of S3-like, indicating that it is dependent on Notch (*Figure 6—figure supplement 1M–O*). Hence, we have identified a region of Notch signalling in the late larval notum that is induced by Dl independently of Mib1 and Neur and probably of ubi altogether.

## Neur can activate Dl independently of ubi

DSL signalling during wing development depends on Mib1. The alternative E3-ligase for Dl, Neur, is only detected in sensory organ precursors, a small late arising minority of cells in the wing pouch. Hence, the experiments presented so far investigated the connection of Mib1 with the Ks in the ICD of Dl. In order to test the connection with Neur, we co-expressed it together with the Dl variants in a *mib1* null mutant background. Expression of Neur alone had no effect on the expression of Wg in normal discs (*Figure 7A*). Co-expression of Neur with Dl-HA in wildtype wing discs induces strong ectopic expression of Wg similar to that already seen with Dl-HA alone (*Figure 7B*, arrows; compare with *Figure 2E*). However, we consistently observe a loss of the anterior stripe of Wg expression in the dorsal half of the pouch (*Figure 7B*, arrowhead). As previously reported, expression of Neur in *mib1* mutants re-established the expression of Wg along the D/V boundary in the area of the *ptc*Gal4 domain (*Le Borgne et al., 2005*; *Wang and Struhl, 2005*) (*Figure 7C*, arrow). The co-expression of Neur and Dl-HA in *mib1* mutant wing discs resulted in induction of an ectopic

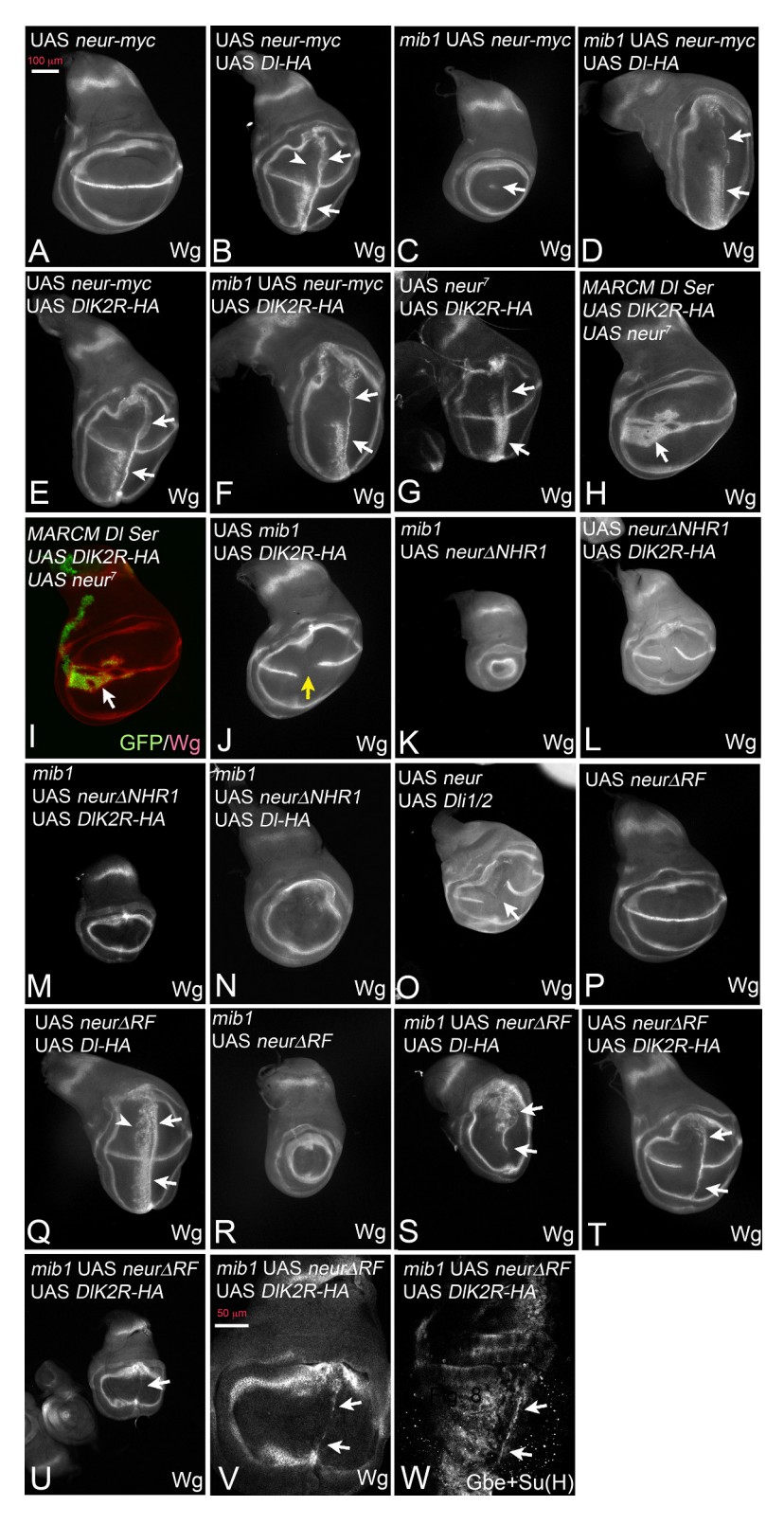

**Figure 7.** Activation of Dl-HA and DlK2R-HA by Neur. With exception of (**H, I**) *ptc*Gal4 was used for expression of the constructs. (**A**) Expression of Neur has no effect on expression of Wg along the D/V boundary. (**B**) Co-expression of Neur and Dl-HA causes a strong ectopic expression of Wg (arrows). The arrowhead points to the dorsal half where the anterior stripe of ectopic Wg expression is suppressed. (**C**) Expression of Neur in *mib1* mutants restores expression of Wg along the D/V boundary (arrow). (**D**) Co-expression of Dl-HA with Neur in *mib1* mutants induces ectopic Wg expression in a

*Figure 7 continued on next page*

*Figure 7 continued*

manner comparable to wildtype (arrows, compare with B). (E, F) Co-expression of Neur with DlK2R-HA in wildtype (E) or *mib1* mutants (F) results in a strong ectopic activation of Wg expression (arrows). (G) Co-expression of an independently generated Neur construct with DlK2R-HA induces a similar ectopic activation of Wg expression (arrows). (H, I) *Dl Ser* mutant MARCM clones (green in I) co-expressing Neur and DlK2R-HA. Wg is ectopically induced in the mutant cells despite the lack of the function of *Dl* and *Ser* (arrow). (J) Co-expression of Mib1 and DlK2R-HA does not induce ectopic Wg expression. The yellow arrow highlights the gap in the expression of Wg along the D/V boundary, which is typical for expression of DlK2R-HA. (K) Expression of NeurΔNHR1 cannot restore the expression of Wg along the D/V boundary. (L) Its co-expression with DlK2R-HA cannot induce ectopic expression of Wg. The phenotype resembles that induced by expression of DlK2R-HA alone (compare with *Figure 2A*). Likewise Co-expression of NeurΔNHR1 with DlK2R-HA (M) or Dl-HA (N) in *mib1* mutants fails to induce ectopic expression of Wg. (O) Co-expression of Neur with Dli1/2, cannot induce ectopic expression of Wg. As in case of expression of Dli1/2 alone, the expression of Wg along the D/V boundary is interrupted (arrow). (P, Q) Expression of NeurΔRF does not affect expression of Wg along the D/V boundary in wildtype discs and co-expression with Dl-HA in the wildtype results in ectopic expression of Wg comparable to expression of Dl-HA alone (Q, arrows, compare with *Figure 2E* and *Figure 1F*). (R) NeurΔRF cannot restore the D/V expression of Wg in *mib1* mutants. (S) Co-expression of NeurΔRF with Dl-HA in *mib1* mutants also induces strong expression of Wg (arrows). Note, that the expression is weaker than in the wildtype disc shown in (Q). (T) Likewise, co-expression of NeurΔRF with DlK2R-HA can induce strong ectopic expression of Wg (arrows). (U–W) Co-expression in *mib1* mutants induces weak ectopic expression of Wg and Gbe + Su(H). (V, W) Magnification of the wing area of the disc shown in (U, arrow). The comparison with (T) reveals that the ectopic expression of Wg is weaker than in wildtype discs. The arrows point to the ectopic expression of Wg (V) and Gbe + Su(H) (W).

DOI: https://doi.org/10.7554/eLife.27346.020

The following figure supplement is available for figure 7:

**Figure supplement 1.** Subcellular localisation of Neur and NeurΔRF.

DOI: https://doi.org/10.7554/eLife.27346.021

expression domain of Wg that was comparable to that observed in wildtype discs (*Figure 7D*, arrows; compare with B).

To our surprise, co-expression with Neur dramatically enhanced also the weak activity of DlK2R-HA in normal and also *mib1* mutant wing discs. Strong ectopic expression of Wg was observed in both genetic backgrounds (*Figure 7E,F*, arrows). We confirmed the enhancement of DlK2R-HA by Neur with an independently generated second chromosome UAS *neur* insertion line (*Lai and Rubin, 2001*) (*Figure 7G*, arrows). Note, that the strength of Wg expression achieved by co-expression of Neur with DlK2R-HA or Dl-HA in *mib1* mutants was comparable (compare *Figure 7D* with F). This suggests that the Ks in the ICD of Dl are of minor importance for its activation by Neur. Moreover, cis-inhibition of DlK2R-HA is reduced to a level comparable to Dl-HA (*Figure 7E*, compare with B). Using MARCM, we found that the activation of the Notch pathway by the co-expression of Neur and DlK2R-HA occurred in the absence of endogenous Dl and Ser (*Figure 7H,I*, arrow). This excludes the possibility that the endogenous ligands are involved in the induction of the ectopic Notch activity. We co-expressed Mib1 together with DlK2R-HA and found that in contrast to Neur, co-expression of Mib1 failed to modulate the activity of DlK2R-HA (*Figure 7J*, yellow arrow, compare with E). Thus, the enhancing effect on DlK2R-HA is a unique property of Neur.

NeurΔNHR1 is a variant that is unable to bind to the ICD of Dl, because it lacks the necessary NHR1 binding domain (*Commisso and Boulianne, 2007*). In contrast to Neur, NeurΔNHR1 failed to rescue the expression of Wg along the D/V boundary in *mib1* mutant discs and did not enhance the signalling activity of DlK2R-HA and Dl-HA in wildtype and mutant discs (*Figure 7K–N*). Moreover, co-expression of Neur did not change the phenotype of Dli1/2, which lacks the Neur binding site in its ICD (*Figure 7O*, compare with *Figure 5A*). Hence, Neur must directly bind to the ICD of Dl-HA and DlK2R-HA to promote their signalling activity and reduce cis-inhibition of DlK2R-HA. The results reveal fundamental differences between Neur and Mib1 in the activation of Dl.

We next asked whether Neur requires its E3-ligase activity for activation of Dl-HA and DlK2R-HA. To do so, we used a variant that lacks the RF, which catalyses the ubi reaction (NeurΔRF) (*Pavlopoulos et al., 2001*). The subcellular localisation of this variant resembles that of Neur, indicating that the RF is not necessary for localisation of Neur (*Yeh et al., 2001*). We have confirmed the correct localisation for the constructs used in our experiments (*Figure 7—figure supplement 1*). Similar to Neur, expression of NeurΔRF alone did not affect expression of Wg along the D/V boundary and did not significantly affect the activity of Dl-HA in wildtype wing discs (*Figure 7P,Q*). However, unlike Neur, it does not abolish the dorsal anterior stripe of ectopic Wg expression observed if Dl-HA is expressed alone (*Figure 7Q*, arrowhead, compare with B). It was also unable to rescue Wg expression in *mib1* mutants (*Figure 7R*). Nevertheless, NeurΔRF strongly enhanced the activity of Dl-

HA in *mib1* mutant discs (*Figure 7S*). In comparison to Neur, a significant difference in the strength of the ectopic expression of Wg was observed between wildtype and *mib1* mutant discs (compare *Figure 7Q,S and B–D*), indicating a requirement of the RF for full activity. However, the result indicates that Neur can activate Dl in a manner independently of catalysing ubi of its ICD or other components of the endocytic machinery. NeurΔRF was also able to enhance the activity of DlK2R in wildtype discs, indicated by the induction of a stripe of ectopic expression of Wg in posterior boundary cells (*Figure 7T*, arrows). A weak stripe was induced if both proteins were expressed even in *mib1* mutant discs (*Figure 7U–W*, arrows). Altogether, the results show that Neur, in contrast to Mib1, can activate Dl in an ubi-independent manner. However, comparison revealed that the induction of Wg expression upon co-expression of Dl and NeurΔRF was stronger than that of DlK2R and NeurΔRF (compare *Figure 7S* with V). In case of Neur the induction of Notch activity was comparable. This suggests that the Ks in the ICD of Dl are important for full activity, especially when Neur lacks its ability to ubiquitylate. The reason for this paradoxical requirement is not clear at the moment. A possible explanation could be a contribution of the Ks to the conformation of the ICD in a manner that cannot be completely replaced by the introduced Rs.

## Ubi-independent Dl signalling is sufficient for specification of neural fates

Since Neur can stimulate Dl activity even in the absence of Ks in its ICD, we tested the activity of the various Dl variants in contexts, where Notch signalling is more dependent on endogenous Neur. We had shown earlier that two such developmental processes are (a) the asymmetric cell division of ganglion mother cells (GMCs) and (b) the selection of the sensory organ precursor (SOP) of macrochetae in the larval notum.

During GMC asymmetric division the daughter cells rely on Dl/Notch signalling to turn on Hey expression in one of the two siblings (*Monastirioti et al., 2010*) (*Figure 8A–B'*). *neur* loss of function severely compromises this signalling, whereas Mib1 only plays an accessory role (*Monastirioti et al., 2010*): 80% of *neur*-clones lose Hey expression (*Figure 8C,C'*), which rises to 100% in a *mib1* background. However, *mib1* in the presence of Neur shows no loss of Hey. Using the MARCM system, we substituted endogenous Dl by DlK2R-HA or Dli1/2 in the developing late larval CNS. The clones were scored in the central brain and ventral nerve cord. As a control, we confirmed that removal of both endogenous ligands results in 96% of the lineages (clones) being Hey negative (*Figure 8D,D'*) and this is fully rescued (0% Hey negative) by UAS Dl-HA, but not Dli1/2 (*Figure 8E–F*). When we expressed DlK2R-HA in *Dl Ser* MARCM clones, we obtained a full rescue of Hey expression; only 4% of lineages were Hey negative (*Figure 8G,G'*). Thus, unlike the wing pouch, where loss of Ks severely compromises Dl activity, in this context DlK2R-HA was able to sustain Notch signalling. Since Neur is needed but Ks are dispensable, we wondered whether the catalytic activity of Neur is necessary. We therefore generated *neur* mutant MARCM clones expressing NeurΔRF. These showed complete rescue of Hey (100% of clones are positive; *Figure 8H,H'*). Taken together with the DlK2R-HA result, this supports a model where ubi is not required for Dl signalling in the context of GMC asymmetric cell division.

Another Neur-dependent Notch signalling context is the selection of the sensory organ precursor (SOP) of the bristle sensillum in the notum. This occurs within proneural clusters, which are defined by the expression of proneural genes by Dl induced Notch signalling (*Modolell and Campuzano, 1998*) (*Figure 9A*). We recently introduced a MARCM based test system to test variants of Dl in this selection process (*Pitsouli and Delidakis, 2005*). MARCM clones double mutant for *Dl* and *Ser* cause the development of clusters of SOPs instead of single ones, termed neurogenic phenotype (*Figure 9A*). This phenotype can be revealed by anti Hindsight (Hnt) staining, which labels mature SOPs (*Figure 9B,C*, arrows). The neurogenic phenotype is completely suppressed by expression of Dl-HA (*Pitsouli and Delidakis, 2005*) (*Figure 9D,E*, arrows). Similarly, DlK2R-HA strongly suppressed the *Dl Ser* mutant neurogenic phenotype (*Figure 9F,G*, arrows). This result suggests that the ubi-independent mechanism is sufficient for selection of most SOPs. To test whether DlK2R-HA requires the function of Neur during the selection process, we repeated the experiments with DlK2R[i1 ala]-HA that lacks the Neur binding site. DlK2R[i1 ala]-HA failed to suppress the neurogenic phenotype in *Dl Ser* double mutant MARCM clones, indicating that direct binding of Neur is required for the selection of the SOP by DlK2R-HA (*Figure 9H,I*, arrows).

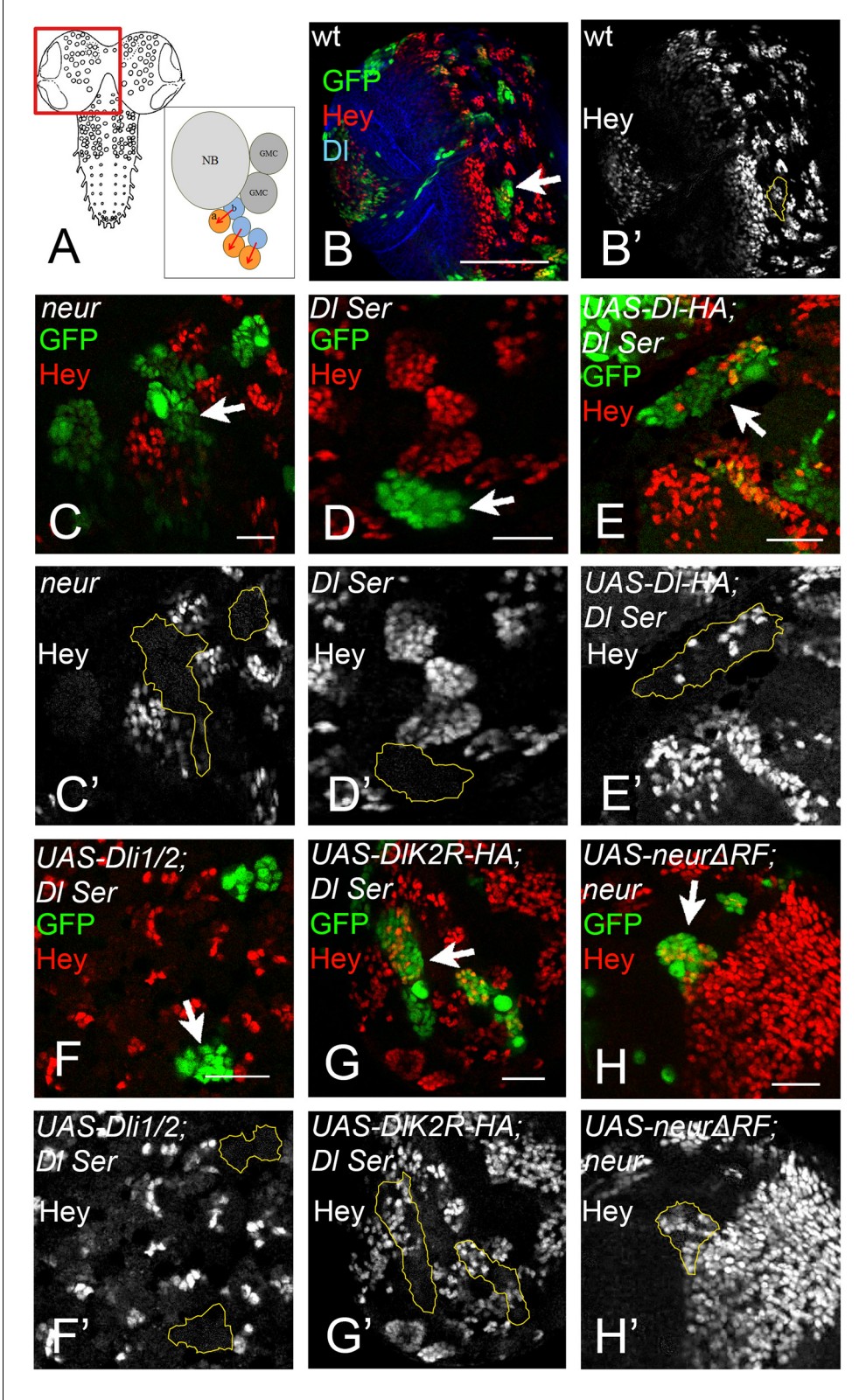

**Figure 8.** (A) Graphic of the larval CNS and a neuroblast lineage. Arrows depict events of Notch signalling. (B) A brain hemisphere (area boxed in A) with clonal lineages in green. One marked lineage is highlighted by the arrow. The large cell nearest to the arrow is the neuroblast. Green: GFP, Red: Hey, Blue: Delta. (C–H) Enlarged areas of brain lobes containing marked clones of various genotypes, highlighted by arrows. In all panels GFP (green) marks

*Figure 8 continued on next page*

*Figure 8 continued*

the mutant lineages. Red: Hey. C′ –H′ show the Hey pattern alone; selected clone borders are drawn in yellow. (**C**) Several GFP marked lineages homozygous for the null allele *neur* (*Glittenberg et al., 2006*). (**D**) A GFP-marked clone double mutant for *Dl* and *Ser*. None of the marked cells express Hey. (**E**) *Dl Ser*; Dl-HA. Wt Dl restores Hey expression in the mutant lineages. (**F**) *Dl Ser*; Dli1/2. No rescue of Hey expression is observed. (**G**) *Dl Ser*; DlK2R-HA. Hey is rescued as efficiently as by wt Dl. (**H**): *neur* (*Glittenberg et al., 2006*); NeurΔR. The catalytically inactive form of Neur rescues Hey expression.

DOI: https://doi.org/10.7554/eLife.27346.022

Lqf binds ubiquitylated cargo by two UIMs. In order to further evaluate the role of ubi of the DSL ligands during SOP selection, we used a "reading "defective variant of Lqf in which UIM1 was mutated and UIM2 deleted (*Xie et al., 2012*). This GFP-tagged variant, termed LqfUIM1$^{3E/3A}$-ΔUIM2-GFP is controlled by the endogenous *lqf* promoter and cannot recognise ubiquitylated cargo (*Xie et al., 2012*). As previously shown, LqfUIM1$^{3E/3A}$-ΔUIM2-GFP partially rescues *lqf* null mutant flies, which normally die during embryogenesis, to the pharate adult stage (*Xie et al., 2012*). The pharate adults displayed severe patterning defects that were similar to those described for *mib1* mutants. The phenotype will be described in detail elsewhere. Importantly, the rescued flies displayed a nearly normal bristle pattern with only the occasional duplicated large bristle and a higher density of small bristles (*Figure 9J–L*, arrowhead in K, L). A similar bristle phenotype was described for *mib1* mutants (*Le Borgne et al., 2005*). The analysis of the wing imaginal discs of the LqfUIM1$^{3E/3A}$-ΔUIM2-GFP rescued *lqf* flies revealed a nearly normal pattern of SOPs, just like in *mib1* mutants (*Figure 9M,O*, compare with A). Altogether, these data confirm that ubi-independent, but Neur dependent DSL signalling prevails during SOP selection. Note, that complete loss of *lqf* function caused a neurogenic phenotype in *mib1* mutant discs (*Figure 9P*, arrow), indicating that Lqf (but not its UIMs) is required for the ubi-independent selection of the SOP.

## Discussion

Previous work established that the activity of the members of the DSL family depends on the function of the E3-ligases Neur and Mib1 that sends them into Lqf/Epsin dependent endocytosis. This activating endocytosis is different from bulk endocytosis, which is - in the case of Dl - not dependent on Mib1 or Lqf (*Le Borgne et al., 2005*; *Wang and Struhl, 2005*). In order to account for these dependencies on E3-ligases, it has been suggested that ubi of the DSL protein ICDs by Neur and Mib1 initiates a special endocytosis event (*Weinmaster and Fischer, 2011*). This event either creates a pulling force that is essential for Notch S2 cleavage and subsequent ecto-domain shedding, or sends the ligands through a recycling pathway where they mature into the active form. However, it is not clear why in a given process only one of the two E3-ligases is required or why Neur strongly affects the endocytosis of Dl, but not of Ser, whereas Mib1 affects the endocytosis of the ligands in the opposite way (*Le Borgne et al., 2005*; *Wang and Struhl, 2005*; *Le Borgne and Schweisguth, 2003*).

Here, we further assessed the role of Ks and ubi during activation of Dl, using different tissues of *Drosophila* as test systems. One main finding of our work is that the activity of Dl has three components: One is dependent on Mib1/Neur/E3-ligase activity and probably ubi, the second is Neur-dependent, but ubi-independent and finally the third is independent of any of these factors. Our analysis suggests that the ubi-independent activities probably account for the residual Notch activity observed in *mib1* null mutants. We identified two normal developmental processes, GMC division and SOP selection, which rely on the ubi-independent/Neur-dependent activity of Dl. The only example of a ubi-independent/Neur-independent case of Dl signalling is the S3 notum region expression of Gbe + Su(H) and the endogenous target gene E(spl)mß. Unfortunately, it is not known in which developmental process this expression domain is involved. The latter ubi-/Neur-independent pathway appears to be restricted to Dl as we did not observe ectopic activation of Gbe + Su(H) upon *Ser* expression in *mib1* mutants.

Recent evolutionary studies suggest that the ubi-independent signalling mode of Dl might be the evolutionary ancient one. *Trichoplax adherens* is the only known member of the basal metazoan phylum Placozoa. It possesses all crucial elements of the Notch pathway members except Mib1 and

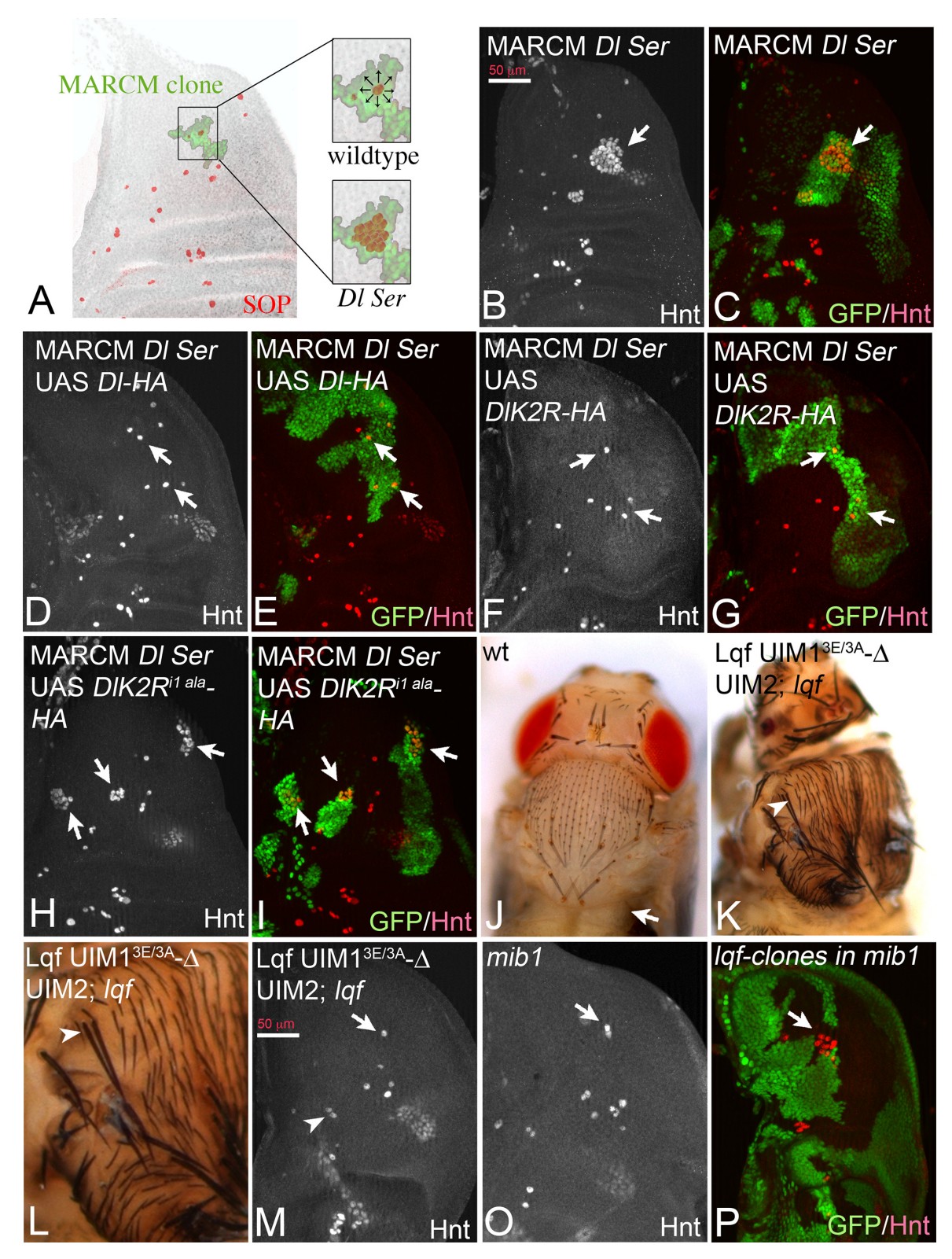

**Figure 9.** Specification of SOPs in an ubi-independent manner. SOPs are revealed by anti Hnt staining. (**A**) Schematic drawing of the distribution of the SOPs of the large bristles in a wildtype disc (red cells). In the wildtype the nascent SOP sends an inhibitory signal via the Notch pathway to prevent SOP development in its neighbours. The loss of the activity of the Notch pathway, e.g. in clones mutant for genes involved in Notch signalling, results in the formation of clusters of SOPs (neurogenic phenotype). (**B–I**) *Dl Ser* double mutant MARCM clones. The SOPs are revealed by Hnt expression. Clones

*Figure 9 continued on next page*

*Figure 9 continued*

are labelled by GFP (green). (B, C) A *Dl Ser* double mutant MARCM clone. The clone includes a proneural cluster (arrow). Many cells have adopted the SOP fate in the absence of *Dl* and *Ser* function (neurogenic phenotype). (D, E) The neurogenic phenotype is suppressed and a single SOP is observed in a *Dl Ser* double mutant MARCM clone that expresses Dl-HA (arrows), indicating that selection of the SOP is restored. (F, G) A similar rescue of the selection process is observed when DlK2R-HA is expressed in *Dl Ser* double mutant MARCM clones (arrows). (H, I) In contrast no rescue is observed when DlK2R[i1 ala] is expressed in the clones (arrows). (J) Notum of a wildtype fly showing the regular bristle pattern. (K, L) Notum of a *lqf* mutant fly rescued by Lqf UIM1[3E/3A]-ΔUIM2-GFP. (L) A magnification of the notal area of the fly shown in (K). An occasional duplication of a large bristle is highlighted by the arrowhead. The density of the smaller bristles is increased compared to the wildtype. However, most bristles are well separated from each other. (M, O) Comparative analysis of SOP formation in wing discs of *lqf* mutant flies rescued by Lqf UIM1[3E/3A]-ΔUIM2-GFP and *mib1* mutant discs. The arrow highlights a SOP at the posterior dorso-central position. A nearly wildtype pattern of SOPs is observed in both genotypes. The arrowhead in (M) points to two adjacent developing SOPs that probably give rise to the bristle duplication observed in the imago (compare with K, L, arrowhead). However, no neurogenic phenotype is observed and the pattern of SOPs of both genotypes is similar. (P) *lqf* mutant clones in the notum of a *mib1* mutant wing disc revealed by absence of GFP (green). The arrow highlights the region where the clone includes a proneural cluster. The loss of the function of *lqf* causes a neurogenic phenotype.

DOI: https://doi.org/10.7554/eLife.27346.023

Neur (*Gazave, 2009*). The E3-ligases appear to be later additions. Moreover, the only ligand present in the *Trichoplax* genome belongs to the Dl family. In the light of our results, it is possible that Notch signalling in *Trichoplax* occurs solely by the probably more ancient ubi-independent mechanism. The Ser-like ligands were probably only introduced later in evolution, after the recruitment of the E3-ligases.

Besides the discovery of the ubi-independent modes, we here confirm that the Ks in the ICD of Dl and Ser are important for their signalling abilities. In addition, our results indicate that the loss of the Ks in Dl also results in a defect in ligand degradation and delayed endocytosis, suggesting the Ks in the ICD are required for the correct degradation. This is in good agreement with our previous finding that the endocytosis of Dli1/2 that lacks the Mib1 and Neur binding sites is delayed (*Daskalaki et al., 2011*). We did not find significant differences in the number of Rab7 positive endosomes that are also positive for Dl-HA or DlK2R-HA, however the total amount of DlK2R-HA on these endosomes was increased compared to Dl-HA. Together with the persistence of DlK2R-HA in the plasma membrane, we favour a model where ICD ubi enhances trafficking of Dl-HA at many stations along the endocytic route and ultimately increases its degradation.

A novel finding is that Neur and Mib1 activate Dl by different mechanisms. In contrast to Mib1, Neur can activate DlK2R-HA to a level that is very similar to that of Dl-HA upon co-expression, also in the absence of *mib1* function. This activation requires direct binding of Neur to the ICD of Dl. This indicates that Neur can strongly activate Dl in a manner independently of ubi of its ICD. The findings are in agreement with the finding that Neur is involved in processes, which appear to depend on ubi-independent Dl signalling, e. g. the selection of neural fates in the brain and the PNS. Previous experiments suggested that Mib1 cannot rescue the embryonic defects of *neur* mutants (*Le Borgne et al., 2005*). It is likely that Mib1's lack of this second, ubi-independent function is the reason for this failure. One possibility for the activation of Dl by Neur might be that it acts as an adapter that connects Dl to the endocytic machinery in the identified developmental processes. In this way Dl could be incorporated into endocytic vesicles without ubi. This adaptor function of Neur is consistent with the observation that it frequently co-traffics with Dl and that expression of Dl can dramatically change the subcellular distribution of Neur (*Daskalaki et al., 2011*; *Skwarek et al., 2007*). Recent work indicates that this adapter function might be more widespread among E3-ligases than anticipated since it has been shown that also the E3-ligase Suppressor of deltex, which plays an important role in endosomal trafficking of Notch, induces the endocytosis of Notch in an ubi-independent manner (*Yamada et al., 2011*).

The selection of the SOP occurs within groups of cells, termed proneural clusters, which are defined by the expression of proneural genes (*Modolell and Campuzano, 1998*). It has previously been shown that this selection occurs within a subgroup of the proneural cluster (*Troost et al., 2015*). It is termed the Neur group, since the cell that first expresses Neur will develop into the SOP and inhibits its neighbours by increased Dl signalling. The requirement for Neur to increase Dl signalling is puzzling since the cells of the subgroup also express Mib1. The discovery of the ubi-independent activation of Dl by Neur provides an explanation why Neur might be required for correct SOP

selection, since it could boost Dl signalling in a different manner than Mib1. Interestingly, our results indicate that also Lqf contributes to SOP selection by an ubi-independent function, as the process is to a large extent rescued by LqfUIM1$^{3E/3A}$-ΔUIM2-GFP in *lqf* mutants. Hence, Lqf might be part of the Neur adapter complex.

Another so far not well-appreciated finding is that the ICD contributes to cis-inhibition. It has previously been shown that the ECD of the ligands is involved in cis-inhibition (*Fleming et al., 2013*; *Glittenberg et al., 2006*; *Li and Baker, 2004*). Here, we found that the cis-inhibitory ability of Dl strongly increases if the Ks in its ICD are lost. We also showed that loss of *mib1* results in a global increase of cis-inhibition by endogenously expressed DSL ligands. Thus, it appears that both parts of Dl are required for cis-inhibition: The ECD appears to execute the necessary physical interactions with Notch, while the ICD adjusts the strength of cis-inhibition via recruiting E3-ligases and promoting ubi, which suppresses cis-inhibition.

How might Dl/Notch cis-inhibition be suppressed by ubi of the ligand ICD? The ubiquitin moiety alone might already promote the separation of the Dl/Notch cis-pair on the cell surface. Alternatively, ubi enables binding of Epsin/Lqf via the UIMs, which separates the pair. The similarity between the phenotype of *mib1* mutants and the LqfUIM1$^{3E/3A}$-ΔUIM2-GFP rescued *lqf* mutants favours the second possibility. Another possibility is that ubi/Lqf sends Dl into the recycling pathway as previously suggested (*Wang and Struhl, 2004*). Separation of the Dl/Notch ECDs is a likely event during this recycling thanks to the acidic pH in the endosome lumen. It is possible that the majority of newly synthesised Dl (and probably Ser) on its way to the cell surface is initially engaged in cis-interaction with Notch. Unless their separation is promoted by ubi, only a minor amount of Dl at the surface would be free to signal. This diminished signalling would justify the reduced (but not eliminated) Notch activity, observed in *mib1* mutants (e. g. S3 expression of Gbe + Su(H)).

We found that co-expression of Neur also reduces the cis-inhibitory effect of DlK2R-HA and at the same time strongly enhances its signalling activity. To do so, Neur must directly bind to the ICD of DlK2R-HA. These results raise the possibility that the binding of Neur might directly separate the Dl/Notch cis-pair or, as a possible endocytic adapter, stimulates endocytic recycling upon which the cis-pair is separated. Future work is needed to clarify whether the cis-pair separation is the only mechanism through which ubi activates DSL signalling, or whether it also contributes to the pulling force which seems to be a prerequisite for Notch-activation in several contexts.

## Materials and methods

### Drosophila stocks and genetics

*mib1$^1$*, *mib1$^2$*, *mib1$^{EY09780}$* (*Le Borgne et al., 2005a*), *lqf$^{L71}$*(*Xie et al., 2012*) and *lqf$^{1227}$* FRT2A (*Wang and Struhl, 2004*), *Psn$^{C2}$* FRT2A (*Struhl and Greenwald, 1999b*), *Dl$^{rev10}$ Ser$^{RX82}$* FRT82B (*Micchelli et al., 1997*), *H$^{E31}$* (*Bang and Posakony, 1992*). Ser- (RRID:BDSC_59284) and Dl-MIMIC-GFP(RRID:BDSC_59819) (*Nagarkar-Jaiswal et al., 2015*). *Neur$^1$* (*Commisso and Boulianne, 2007*). All used alleles are null alleles.

UAS constructs: UAS *neur* (*Weinmaster and Fischer, 2011*; *Lai et al., 2001*), UAS *neur-myc* (third chromosome) and UAS *neurΔRF-GFP* (*Pavlopoulos et al., 2001*), UAS *neurΔNHR1-V5* (*Commisso and Boulianne, 2007*), UAS *N-LV* (*Loewer et al., 2004*), UAS *mib1* (*Lai et al., 2005*), UAS *N-RNAi* (RRID:BDSC_7078), UAS *Dli1* and UAS *Dli1/2* (*Daskalaki et al., 2011*), UAS *Dl$^{stu}$* (a gift of D. Horowicz and D. Henrique).

Further stocks: *ptc*Gal4 (RRID:BDSC_2017), *tub*Rab7-YFP (*Marois et al., 2006*), Gbe+Su(H)-lacZ (*Furriols and Bray, 2001*), *tub*Gal4 *tub*Gal80$^{ts}$ (RRID:BDSC_5138, RRID:BDSC_7018), *ci*Gal4 (*Croker et al., 2006*) and *vg*BE-GFP (*Zecca and Struhl, 2007*).

### Generation of constructs

Dl-HA was generated by introduction of a synthesized fragment from the NdeI restriction site of the ICD onwards. The DlK2R-HA was generated by replacing the ICD of Dl-HA by a synthesised ICD in which all lysines are replaced by arginines. Constructs were cloned in pUAST *attB* and inserted into the landing site at 51C. Gene synthesis was performed by GenScript. DlK2R-HA was used to generate DlK2R$^{i1\ ala}$-HA (NEQNAV to A, see [*Fontana and Posakony, 2009*]) and Dl$^{i2\ ala}$-HA (IKNTWDK to A) by site directed mutagenesis. All constructs were sequenced before injection and subsequently

inserted into the *attP* landing site at 51C. The LqfUIM1$^{3E/3A}$-ΔUIM2-GFP *attB* construct was obtained from J. Fischer (described in [*Xie et al., 2012*]) and inserted into the *attP22A* landing site. Transgenesis was partly performed by BestGene Inc. The necessary primers were purchased from Sigma-Aldrich.

## Clonal analysis
MARCM and conventional clones were induced during the first larval instar stage (24–48 hr after egg laying).

## Western blot analysis
Wing imaginal discs were dissected from third instar larvae and boiled in Laemmli buffer at 95°C for 10 min. Each lysate contained at least 10 discs. Proteins were separated by SDS-PAGE gel electrophoresis and blotted according to standard protocols. Antibodies used: HA (anti-HA, High Affinity, Roche (3F10) RRID:AB_390919), alpha-Tubulin (Sigma-Aldrich T5168, RRID:AB_477579), HRP-conjugated secondary antibodies were purchased from Jackson Immuno Research.

## Ubiquitylation of Dl-HA and DlK2R-HA
### Transfections
S2-DGRC cells (RRID: CVCL_Z992) were transfected with the calcium phosphate precipitation method. The cell line was obtained from Drosophila Genomics Resource Center, supported by NIH grant 2P40OD010949. Its identity was confirmed by visual inspection of the cell morphology and its growth kinetics in M3 medium (SIGMA-ALDRICH S8398) supplemented with 10% Fetal Bovine Serum (GIBCO 10270). A mycoplasma test is usually not done for S2 cells.

For each sample $10^7$ cells were used (10 cm dish). The medium was supplemented with biotin (50 µM) for 24 hr and the lysosomal inhibitor E64 for 5–6 hr. Finally, the cells were collected 96 hr post-transfection. For each transfection, the following amounts of plasmids were used: 5 µg of pIZDeltaHA.His(6) or pIZDeltaK2R.HA.His(6)(this work), 2.5 µg pAC5-Gal4 (Addgene #24344), 2.5 µg of Actin5C-bioUbi(6)-BirA-puro or Actin5C-BirA-puro and 5 µg of pUAST-EGFP:Neuralized or pUAST-Mindbomb1:HisMyc or in the case of the negative controls pUAST-attB as a stuffer plasmid.

### Lysis and neutravidin pull-down
The cells were washed 3 times in 1X PBS to remove excess of free biotin. The cells were resuspended in lysis buffer (0.5 ml/10 cm dish; 8 M urea, 1% SDS, 50 mM N-ethylmaleimide, 1 × protease inhibitor cocktail (Roche) in PBS, and sonicated (4 cycles 30 s ON, 30 s OFF). For the pull-down of the total biotinylated and ubiquitylated proteins, cell extracts were incubated with 50 ul of clear bed volume of high-capacity NeutrAvidin-agarose beads (Thermo Scientific, Waltham, Massachusetts, USA) overnight at room temperature. Washes were done according to Franco et al.: 2 × WB1, 3 × WB2, 1 × WB3, 3 × WB4, 1 × WB1, 1 × WB5 and 3 × WB6 [WB1: 8 M urea, 0.25% SDS in PBS; WB2: 6 M guanidine hydrochloride in PBS; WB3: 6.4 M urea, 1 M NaCl, 0.2% SDS in PBS (pre-warmed to 37°C); WB4: 4 M urea, 1 M NaCl, 10% isopropanol, 10% ethanol, 0.2% SDS in PBS; WB5: 8 M urea, 1% SDS in PBS; WB6: 2% SDS in PBS]. Samples were eluted in 100 µl of 4 × Laemmli sample buffer with 100 mM DTT by two cycles of heating (5 min; 99°C), with vortexing in between. References: (*Pirone et al., 2017*; *Franco et al., 2011*)

## Immunostaining and microscopy
Antibody staining was performed according to standard protocols (*Klein, 2006*). Antibodies used: anti-Wg antibody (DSHB Iowa RRID:AB_528512), anti-ß-Gal (Cappel/MP Biomedicals RRID:AB_2313831), anti-Rab7 (*Tanaka and Nakamura, 2008*) (RRID:AB_2569808). Fluorochrome-conjugated secondary antibodies were purchased from Invitrogen/Molecular Probes. Images were obtained with a Zeiss AxioImager Z1 Microscope equipped with a Zeiss Apotome or a Leica SP6 confocal microscope.

## Imaris image analysis for Dl-HA and DlK2R-HA in Rab7 positive vesicles
For quantitative analysis of Dl-HA/DlK2R-HA in Rab7 positive vesicles, the image analysis software 'Imaris' (RRID:SCR_007370, Bitplane, Zurich, Switzerland) was used. Z-Stacks of imaginal discs were

acquired with a Zeiss AxioImager Z1 Microscope equipped with a Zeiss Apotome, applying the same microscopy hardware settings (e.g. exposure time) to ensure reproducibility between datasets. Individual Rab7 vesicle volumes were identified utilizing the 'spot feature' of *Imaris* with an initial size of 0.4 μm. Regrowing of volume, based on the local contrast diameter threshold was allowed later on. A quality threshold was placed on resulting spots to minimise background volumes. For further analysis only vesicles were included, which contain at least voxels with a Dl-HA/DlK2R-HA signal intensity threshold of 1. Finally, the absolute cargo of Dl-HA/DlK2R-HA signal in Rab7 vesicles was defined as the median of the absolute signal sum per vesicle. The average of these values over several biological replicates is displayed in the graph. Statistics were applied as described above to calculate significance levels. The procedure of vesicle volume definition, thresholding and downstream analysis was the same for all datasets. In total 19373 (Dl-HA) and 15720 (DlK2R-HA) vesicles of 3 discs for each genotype were included in the analysis.

## Acknowledgements

We thank Sylvia Tannebaum for excellent technical support. We thank Alina Airich, Verena Behnke, Jessica Nicolai and Sven Fleischer for help in some experiments. We thank G Boullianne, S Bray, S Eaton, M Fortini, E Lai, J Posakony, F Schweisguth, J Fischer and G Struhl for supplying fly stocks and/or reagents used in this work. We thank the Center for Advanced Imaging (CAi) of the Heinrich-Heine-University for expertise and support with microscopy. The Bloomington Stock Center, VDRC and the Developmental Studies Hybridoma Bank supplied fly stocks and reagents. The work of the TK lab and SH was funded by the DFG through KL 1028/3–1 and in part by SFB 1208. Work in the CD lab was funded by the Thalis and Aristeia II programmes from the Greek General Secretariat for Research and Technology and co-funded by the European Social Fund and national resources. Part of the work was funded by the COST Action PROTEOSTASIS (BM1307), supported by COST (European Cooperation in Science and Technology). Special thanks go to Rosa Barrio and Jim Sutherland (CIC Biogune, Bizkaia, Spain) for hosting KK, training her in ubi detection methods and sharing crucial materials prior to publication.

## Additional information

### Funding

| Funder | Grant reference number | Author |
|---|---|---|
| European Cooperation in Science and Technology | BM1307 | Christos Delidakis |
| General Secretariat for Research and Technology | Aristeia II, No 4436 | Christos Delidakis |
| Deutsche Forschungsgemeinschaft | KL 1028/3-1 | Thomas Klein |
| Deutsche Forschungsgemeinschaft | SFB 1208 Teilprojekt B01 | Thomas Klein |

The funders had no role in study design, data collection and interpretation, or the decision to submit the work for publication.

### Author contributions

Nicole Berndt, Ekaterina Seib, Christos Delidakis, Validation, Investigation, Methodology, Writing—review and editing; Soya Kim, Validation, Investigation, Methodology; Tobias Troost, Investigation, Methodology, Writing—review and editing; Marvin Lyga, Formal analysis, Validation, Investigation, Methodology; Jessica Langenbach, Data curation, Formal analysis, Investigation; Sebastian Haensch, Data curation, Software, Validation, Writing—original draft; Konstantina Kalodimou, Investigation, Methodology; Thomas Klein, Conceptualization, Data curation, Formal analysis, Supervision, Funding acquisition, Writing—original draft, Project administration, Writing—review and editing

## Author ORCIDs

Thomas Klein http://orcid.org/0000-0002-2719-9617

## Decision letter and Author response

Decision letter https://doi.org/10.7554/eLife.27346.025
Author response https://doi.org/10.7554/eLife.27346.026

## Additional files

### Supplementary files

• Transparent reporting form
DOI: https://doi.org/10.7554/eLife.27346.024

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
