## [Decision Letter]

Thank you for submitting your article "Ubiquitylation-independent activation of Notch signalling by Delta" for consideration by *eLife*. Your article has been reviewed by three peer reviewers, and the evaluation has been overseen by K VijayRaghavan as the Senior Editor and Reviewing Editor. The following individual involved in review of your submission has agreed to reveal his identity: Alfonso Martine Arias (Reviewer #1).

The reviewers have discussed the reviews with one another and the Reviewing Editor has drafted this decision to help you prepare a revised submission. As you can see, we are very positive about your paper and are sure you will be able to address all major comments speedily.

Summary:

The mechanism of Notch signalling, specifically the mechanics of the trigger, is still a matter of debate. One of the reasons for this is a lack of understanding of the details that lead Delta to interact productively with Notch. Genetic studies, initially from *Drosophila* with important contributions from vertebrates, have implicated special endocytosis events in this mechanism and, as an important element of this process, the ubiquitylation of Delta. Mib and Neur are two ubiquitin ligases implicated in this process through a direct interaction with Delta. Mib is a general ligase whereas Neur is involved in specific process.

There are at least three fundamental findings in this manuscript that will have a strong impact in the Notch field:

i) Delta (but not Serrate) can signal in a ubiquitination independent manner. Furthermore that there are specific processes that use this signalling event, notably the specification of neural progenitors. The authors show two modes of ubiquitylation-independent activation: A weak activation in the absence of both *Mib1* and Neur and a relatively strong activation by Neur-dependent ubiquitylation-independent process.

ii) The ubiquitin ligase Neuralized activates Delta, but does it independent of Delta ubiquitylation itself.

iii) The key role of Delta ubiquitylation is to release the cis-inhibition of the ligand by the receptor, which is then competent for trans-activation in neighbouring cells.

The experiments are well done, the conclusions well supported and the work shows the value of *Drosophila* in eliciting mechanism. There is much detail here that bridges cell biology and pattern formation and the work has the potential to inspire and guide the interpretation of Delta activity in vertebrates.

In addition to providing these fundamental concepts to the field, this work represents a tour de force, in which a plethora of original assays and reagents are now made available to the community. This manuscript is therefore well suited for publication in *eLife*.

If the manuscript has a drawback it is that, in places, it is long. We would suggest that the authors make an effort to shorten it and be a bit more direct. The Discussion in particular might not need the length of some of the arguments.

Essential revisions:

1) It should be made clear in the Abstract and in the Discussion that there are two modes of ubi-independent Notch activation ((5) and (14) above). The first part of the manuscript (Figure 1–Figure 6)) deals with mode (5), and the second part of the manuscript (Figure 7–Figure 9) deal with the second mode.

2) The authors brought evidence that Neur-dependent, ubi-independent Notch signaling (mode 2) occurs in endogenous situations. Is there evidence for endogenous situations where Notch signaling occurs in the absence of both Neur and *Mib1* (mode 1)?

3) The Neur dependent ubi-independent mode is quite intriguing. It suggests that the physical binding of Neur, and not its catalytic activity, is somehow sufficient for DSL activation. Can the author discuss/suggest potential mechanisms for that?

4) In Figure 3 the authors argue that DlK2R-HA (lysin depleted ligand) and Dl-HA can both activate in the absence of *mib1* (and more strongly so in UAS-N background). However, there are still some differences between the phenotypes of the two variants. These differences could arise from (5) the effect of endogenous Dl, or (14) that binding of *Mib1* can affect Dl regardless of its catalytic activity (in the same manner that Neur does later). Hence, the authors should check if in the absence of endogenous Dl, DlK2R activity is independent of *Mib1*. If not, this would indicate that *Mib1* binding to Dl also confer some activity to Dl.

5) The difference between Neur and NeurDelRF (Figure 7) could be explained by improper membrane localization of the mutant. Can the authors show a high-resolution image to verify that both variants are similarly localized to the membrane?

6) The paper is somewhat hard to read. This is mainly because it includes many details. While we appreciate the rigour of the work and thoughtful controls, it seems that some of the details can be taken out (not necessary for proving the point) or at least removed to supplementary figures. Some of the details which are redundant are the showing of both the experiments performed in random chromosomal integration positions (Figure 3—figure supplement 1) and in position 51C. The part about the effect of cell death is quite confusing and interferes with the flow of the paper. Also, the Dll1-Dl variant seems less relevant here (although it is an interesting observation). Another less relevant control (in my mind) is Figure 3. There are probably more panels that can be moved to supplementary and the authors could make an effort to leave only the panels that convey the main point.

In addition There are a few suggestions to improve readability:a) Use schematics to explain the wt and mutant patterns in the wing and notum (similar to what is done in Figure 8).

b) Make sure to explain the abbreviations and jargon.

7) The paper relies mostly on images. There are some cases where the paper would benefit from quantification. These include quantifying% of penetrance for penetrant phenotypes (Figure 2, Figure 3) and measuring the gap formed by cis inhibition for different mutants (Figure 5 vs. Figure 3).

---

## [Author Response]

Essential revisions:1) It should be made clear in the Abstract and in the Discussion that there are two modes of ubi-independent Notch activation ((1) and (2) above). The first part of the manuscript (Figure 1–Figure 6)) deals with mode (1), and the second part of the manuscript (Figure 7–Figure 9) deal with the second mode.

We have re-written the manuscript and made it clear that two modes operate.

2) The authors brought evidence that Neur-dependent, ubi-independent Notch signaling (mode 2) occurs in endogenous situations. Is there evidence for endogenous situations where Notch signaling occurs in the absence of both Neur and Mib1 (mode 1)?

We found that Stripe3 (S3) of the Notch reporter Gbe+Su(H) and the stripe3-like expression domain of E(spl)mß are induced in this mode. We have now added experiments that the S3 is independent of Neur by inducing Neur clones in *mib1* mutants! The results are described in the section termed “A domain of Dl/Notch signalling that is independent of *Mib1* and Neur”. Unfortunately, we do not know what structure arises from this region or what process is running in this region.

3) The Neur dependent ubi-independent mode is quite intriguing. It suggests that the physical binding of Neur, and not its catalytic activity, is somehow sufficient for DSL activation. Can the author discuss/suggest potential mechanisms for that?

Yes, we have included some speculation in the Discussion that Neur might act as a adapter connecting Dl to the endocytic machinery. Alternatively, but mutual exclusive, the binding of Neur separates the Dl/Notch cis pair to free Dl for trans-signalling.

4) In Figure 3 the authors argue that DlK2R-HA (lysin depleted ligand) and Dl-HA can both activate in the absence of mib1 (and more strongly so in UAS-N background). However, there are still some differences between the phenotypes of the two variants. These differences could arise from (1) the effect of endogenous Dl, or (2) that binding of Mib1 can affect Dl regardless of its catalytic activity (in the same manner that Neur does later). Hence, the authors should check if in the absence of endogenous Dl, DlK2R activity is independent of Mib1. If not, this would indicate that Mib1 binding to Dl also confer some activity to Dl.

We have added the necessary experiments: 1) We induced clones double mutant for dl and Ser in *mib1* mutant discs that express DlK2R-HA and found that DlK2R-HA can induce the expression of Gbe+Su(H) in the Dl Ser mutant area (Figure 3). Hence, DlK2R-HA activity is independent of *Mib1* also in the absence of endogenous Dl. In addition we have included the results of experiments where we expressed Dl and DlK2R-HA in *mib1* mutant and looked at the expression of Dl-MIMIC-GFP to see the reaction of the endogenous Dl promoter (Figure 3—figure supplement 4. We found that the induction of expression of Dl by expression of exogenous Dl is severely reduced. We think that the essential loss of the DS-loop results in the observed reduction of Notch activity.

5) The difference between Neur and NeurDelRF (Figure 7) could be explained by improper membrane localization of the mutant. Can the authors show a high-resolution image to verify that both variants are similarly localized to the membrane?

We made high resolution images that show that both variants are localised correctly. These are shown in Figure 7—figure supplement 1. The correct localisation of NeurΔR has already been reported by (Yeh et al., 2001). We have added this in the manuscript.

6) The paper is somewhat hard to read. This is mainly because it includes many details. While we appreciate the rigour of the work and thoughtful controls, it seems that some of the details can be taken out (not necessary for proving the point) or at least removed to supplementary figures.

We have extensively re-written the manuscript along the line suggested by the reviewers and think that it now reads more fluently and the Discussion is shorter (old: 1598 words, new: 1391 words).

Some of the details which are redundant are the showing of both the experiments performed in random chromosomal integration positions (Figure 3—figure supplement 1) and in position 51C.

We removed a half of experiments with the randomly inserted Dl variants. We also re-wrote the corresponding section starting with the experiments with the 51C inserted DlK2R-HA.

The part about the effect of cell death is quite confusing and interferes with the flow of the paper.

We removed this part.

Also, the Dll1-Dl variant seems less relevant here (although it is an interesting observation). Another less relevant control (in my mind) is Figure 3. There are probably more panels that can be moved to supplementary and the authors could make an effort to leave only the panels that convey the main point.

We have moved many of the controls to supplementary figures or removed them completely as suggested by the reviewers.

In addition There are a few suggestions to improve readability:a) Use schematics to explain the wt and mutant patterns in the wing and notum (similar to what is done in Figure 8).

We have included schematics in Figure 1, Figure 2, Figure 3, Figure 6 and Figure 9 which we hope will facilitate the understanding of the experiments.

b) Make sure to explain the abbreviations and jargon.

Hopefully done.

7) The paper relies mostly on images. There are some cases where the paper would benefit from quantification. These include quantifying% of penetrance for penetrant phenotypes (Figure 2, Figure 3) and measuring the gap formed by cis inhibition for different mutants (Figure 5 vs. Figure 3).

We have done a quantification of the degree of penetrance of the non-penetrant phenotypes.

To quantify the increase of the cis-inhibition, we have counted the distance between the two residual stripes of Wg expression along the D/V boundary. However, the results-although very clear – are probably incorrect because of the scar formation in the case of DLK2R-HA and Dl i1/2. Many cells die and the distance is therefore probably even larger.